# Cross-linked polyaniline for production of long lifespan aqueous iron‖organic batteries with electrochromic properties

Haiming Lv[1,2,6], Zhiquan Wei [3,6], Cuiping Han [4] ✉, Xiaolong Yang[1], Zijie Tang[1], Yantu Zhang[2], Chunyi Zhi [1,3] ✉ & Hongfei Li[1,5] ✉

Aqueous iron batteries are appealing candidates for large-scale energy storage due to their safety and low-cost aspects. However, the development of aqueous Fe batteries is hindered by their inadequate long-term cycling stability. Here, we propose the synthesis and application as positive electrode active material of cross-linked polyaniline (C-PANI). We use melamine as the cross-linker to improve the electronical conductivity and electrochemical stability of the C-PANI. Indeed, when the C-PANI is tested in combination with a Fe metal negative electrode and 1 M iron trifluoromethanesulfonate ($Fe(TOF)_2$) electrolyte solution, the coin cell can deliver a specific capacity of about 110 mAh g$^{-1}$ and an average discharge voltage of 0.55 V after 39,000 cycles at 25 A g$^{-1}$ with a test temperature of 28 °C ± 1 °C. Furthermore, mechanistic studies suggest that $Fe^{2+}$ ions are bonded to TOF$^-$ anions to form positively charged complexes $Fe(TOF)^+$, which are stored with protons in the C-PANI electrode structures. Finally, we also demonstrate the use of C-PANI in combination with a polymeric hydrogel electrolyte to produce a flexible reflective electrochromic lab-scale iron battery prototype.

Aqueous rechargeable batteries that are intrinsically safe and inexpensive are promising for large-scale energy storage[1–4]. Iron is among the most abundant elements on earth and is one of the least expensive metals that support modern society. More importantly, iron metal shows good electrochemical performance, i.e., a low redox potential (−0.44 V vs. standard hydrogen electrode) and high theoretical capacity (7557 mAh cm$^{-3}$ or 960 mAh g$^{-1}$); thus, iron metal is emerging as a viable negative electrode active material for rechargeable aqueous batteries[5–7].

However, while monovalent Li$^+$ and divalent Fe$^{2+}$ ion have a similar ionic radius (0.078 nm for Fe$^{2+}$ vs. 0.076 nm for Li$^+$), divalent Fe$^{2+}$ ion carries twice the amount of charge; thus, strong electrostatic interactions occur between the iron ions and the hosting lattice[8–10], resulting in sluggish diffusion and poor cyclic stability[5,6,11]. Currently, research on iron ion batteries remains in a primary stage. The development of stable cathode materials remains a challenge, and the iron ion storage mechanism is still unclear. Only limited cathode materials for iron ion batteries have been reported, such as sulfur[5], Prussian blue analogue[6], VOPO$_4$. H$_2$O[7] and I$_2$[11], and their performances are far from satisfactory in terms of specific capacity and cycling stability (generally <1000 cycles). Therefore, an alternative positive electrode active materials with improved iron storage performance is desirable for the development of aqueous iron-based secondary batteries.

[1]Songshan Lake Materials Laboratory, Dongguan, 523808 Guangdong, China. [2]Key Laboratory of New Energy & New Functional Materials, Shaanxi Key Laboratory of Chemical Reaction Engineering, College of Chemistry and Chemical Engineering, Yan'an University, 716000 Yan'an, Shaanxi, China. [3]Department of Materials Science and Engineering, City University of Hong Kong, Hong Kong 999077, China. [4]Faculty of Materials Science and Engineering, Low Dimensional Energy Materials Research Center, Shenzhen Institutes of Advanced Technology, Chinese Academy of Sciences, 518055 Shenzhen, China. [5]School of System Design and Intelligent Manufacturing, Southern University of Science and Technology, Shenzhen, 518055 Guangdong, China. [6]These authors contributed equally: Haiming Lv, Zhiquan Wei. ✉e-mail: cp.han@siat.ac.cn; cy.zhi@cityu.edu.hk; lihf@sustech.edu.cn

Conductive polymers, such as polyaniline (PANI), have been considered as possible electrode active materials for energy storage systems due to their high electrical conductivity, large theoretical specific capacity[12,13], fast reaction kinetics in the coordination-based redox mechanism[14], and eco-friendly nature[15,16]. More importantly, in contrast to inorganic electrode materials, conductive polymers exhibit a more flexible solid structure and low repulsion for multivalent ion store due to their long-range conjugated aromatic ring. However, conventional PANI suffers from severe electrochemical degradation[17,18], resulting in poor cycling stability of PANI-based batteries.

Herein, we developed an aqueous Fe‖organic battery that combines a cross-linked PANI (C-PANI) cathode, an Fe metal anode, and an aqueous iron (II) trifluoromethanesulfonate (Fe(CF$_3$SO$_3$)$_2$, Fe(TOF)$_2$) electrolyte solution. Compared to conventional PANI, the stability of C-PANI is enhanced due to the implanted crosslinker; therefore, a 39,000-cycle lifespan at 25 A g$^{-1}$ and 28 °C was achieved, as well as an improved rate capability. Moreover, the battery presents high specific

capacity and high rate performance (133 mAh g$^{-1}$ at 25 A g$^{-1}$). We also demonstrated that during the charging and discharging process, Fe$^{2+}$ ions are bonded to TOF$^-$ to form Fe(TOF)$^+$ complex ions, which have lower electrostatic interactions than those of divalent Fe$^{2+}$ ions. Furthermore, a costorage mechanism of Fe(TOF)$^+$ complex ions and protons in C-PANI was proven. Finally, a flexible iron‖C-PANI electrochromic battery was assembled to integrate both electrochromic function and energy storage capability.

## Results

### Synthesis and characterization of PANI and C-PANI

C-PANI was synthesized using an in situ chemical oxidative polymerization technique (Fig. 1a). Ammonium persulfate was added into a mixted solution of aniline, diphenylamine, melamine, and hydrochloric acid. Then, the reaction was slowly stirred at 0–4 °C for over 24 h until the green colour of the conductive emeraldine state appeared[19,20]. During this polymerization process, melamine served as the crosslinker and was grafted between the PANI chains through free

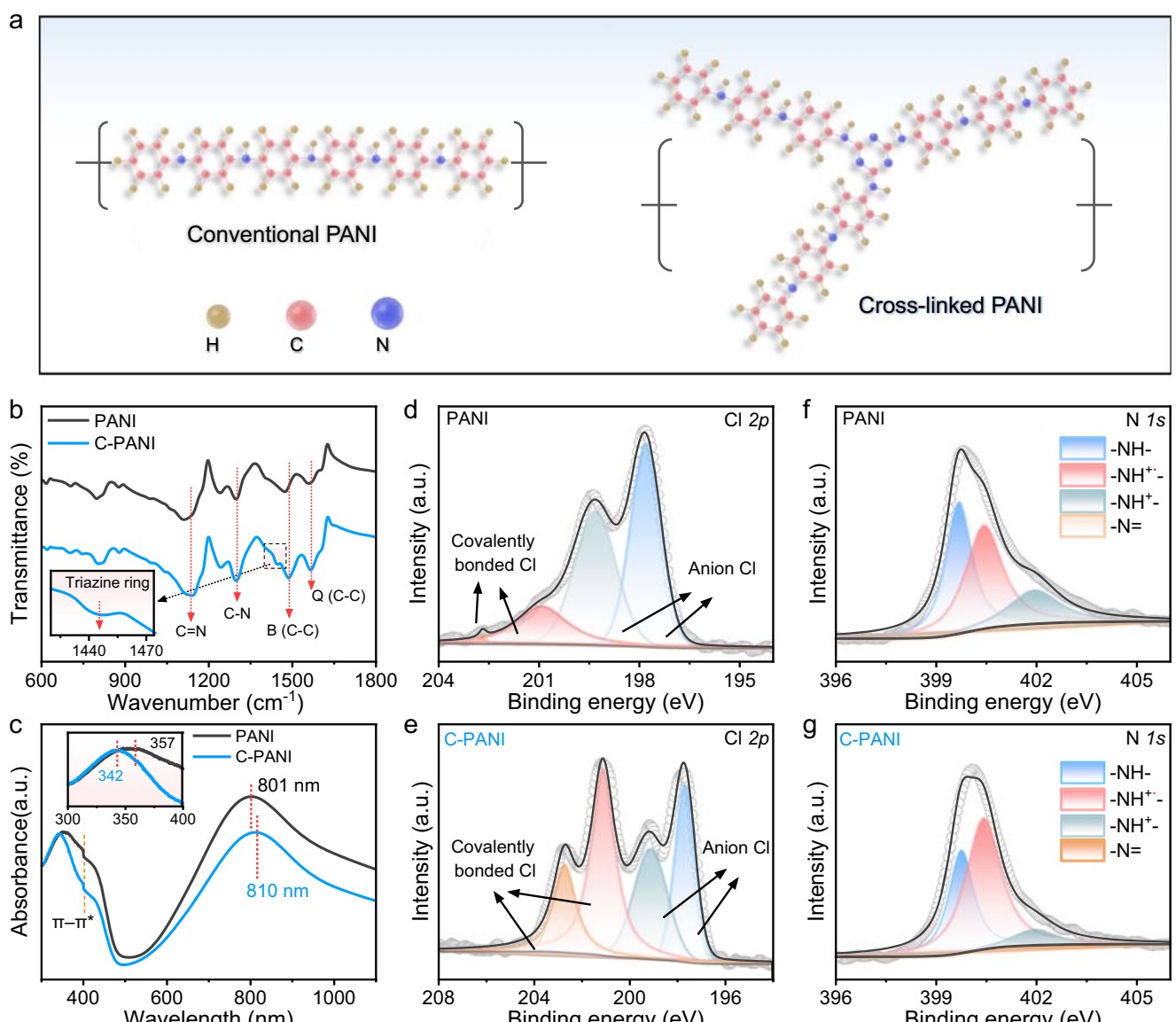

**Fig. 1 | Synthetic routes and spectral characterization of PANI and C-PANI.** **a** Schematic synthesis of conventional PANI and C-PANI. **b** FTIR of conventional PANI and C-PANI. **c** UV–Vis spectra of the conventional PANI and C-PANI samples. High-resolution XPS spectra of the Cl 2*p* core level spectra of PANI (**d**) and C-PANI (**e**). High-resolution XPS spectra: N 1*s* core level spectra of PANI (**f**) and C-PANI (**g**).

radical polymerization of monomers, which promotes electron transport between chains of the PANI. Then, carbon cloth (Supplementary Fig. 1) or gold-deposited nylon 66 film was immersed into the mixture solutions to prepare the C-PANI cathode. For comparison, conventional PANI was synthesized using the same procedure without melamine. The morphology of C-PANI was investigated by scanning electron microscopy (SEM) and transmission electron microscopy (TEM). SEM images of C-PANI exhibit a nanoisland structure on the carbon cloth (inset of Supplementary Fig. 1). The TEM image shows an amorphous feature (Supplementary Fig. 2a), and the corresponding energy-dispersive X-ray spectroscopy (EDS) in Supplementary Fig. 2b–d reveals that C, N, and Cl are uniformly distributed in C-PANI.

To investigate the structural, compositional and functional group information, Fourier transform infrared (FTIR), UV–vis, and Raman spectroscopy measurements were carried out (Fig. 1b, c, and Supplementary Fig. 3). The FTIR spectrum of PANI exhibits characteristic absorption bands at 798 cm$^{-1}$ (aromatic ring and out-of-plane deformation vibrations), 1139 cm$^{-1}$ (vibration of the δ (C − H) structure) 1298 cm$^{-1}$ (C−N$^+$stretching vibration in the polaron structure), 1473 cm$^{-1}$ (benzenoid ring-stretching vibration), and 1560 cm$^{-1}$ (quinonoid ring-stretching vibrations), which are the typical absorption bands of PANI[21–25]. C-PANI produces a different spectrum from that of conventional PANI. For example, the main bands at 1560 cm$^{-1}$, 1473 cm$^{-1}$, 1240 cm$^{-1}$ and 798 cm$^{-1}$ are slightly shifted to 1567 cm$^{-1}$, 1487 cm$^{-1}$, 1244 cm$^{-1}$, and 803 cm$^{-1}$, resulting from chemical environment changes after melamine grafting. Additionally, the triazine ring stretching of the 1444 cm$^{-1}$ band appeared[26], which further suggests that melamine has been implanted. Similarly, the Raman peaks of C-PANI at 1500 cm$^{-1}$ (quinonoid C = N stretching vibration), 1340 cm$^{-1}$(C-N$^+$), and 1167 cm$^{-1}$ (semiminoonoid C-H stretching vibration) are shifted from those of PANI at 1507 cm$^{-1}$, 1329 cm$^{-1}$, and 1172 cm$^{-1}$ (Supplementary Fig. 2)[27–32]. The redshift of the band from 1329 cm$^{-1}$ to 1340 cm$^{-1}$ can be explained by the increase in the chain length and enhanced delocalization of π-electrons along with the conjugated system, which increase the electrical conductivity of C-PANI. The UV−vis absorption spectra are shown in Fig. 1c. These two samples show three absorptions, including ~306–360 nm for the π–π* transition of the benzenoid rings, ~402–412 nm for the polaron–π* transition and ~800–835 nmfor the π–polaron transition, which belong to typical PANI absorption peaks[29]. The implantation of melamine changes the energy band absorption; for example, 357 nm and 801 nm for PANI shifts to 342 nm and 810 nm for C-PANI, as shown in Fig. 1c.

We also employed X-ray photoelectron spectroscopy (XPS) to study the chemical states of PANI and C-PANI, as shown in Supplementary Fig. 4. The high-resolution Cl 2$p$ spectrum was deconvoluted into four peaks, as shown in Fig. 1d, e. The binding energys of Cl 2$p$1/2 and 2$p$3/2 are located at 197.8 eV and 199.3 eV, respectively, which are attributed to chloride ions. The other two peaks at 201.1 eV and 202.7 eV are assigned to covalent chlorine[33–36]. The C-PANI shows 55.9% covalently bonded Cl, while the PANI indicates approximately 18.2% covalently bonded Cl, suggesting that C-PANI exhibits a high conductivity[33]. This was further confirmed by the electrical conductivity test. As shown in Supplementary Fig. 5, each sample was measured 10 times, and the average value was obtained. The average conductivity of C-PANI is 0.38 S cm$^{-1}$, which is higher than the 0.003 S cm$^{-1}$ of PANI. Moreover, compared with that of conventional PANI, the high-resolution N 1$s$ spectrum of C-PANI exhibits more −NH$^+$− (the semiquinone cationic radical at 400.4 eV) (Fig. 1f, g), which is in accordance with the Raman results.

## Electrochemical performance analysis

To test the electrochemical performance of C-PANI, a coin cell was assembled by pairing with an Fe metal anode and a 1 M Fe(TOF)$_2$ electrolyte. The CF$_3$SO$_3^-$ ions in the Fe(TOF)$_2$ electrolyte possess weaker interactions with H$_2$O molecules/metal ions than those of other anions (SO$_4^{2-}$, NO$_3^-$, Cl$^-$) in aqueous batteries[37]; this property is favourable for the desolvation process, resulting in fast kinetics and high Coulombic efficiency. Investigations on the 1 M Fe(TOF)$_2$ electrolyte and anode stability are shown in Supplementary Figs. 6–10. The results of the symmetrical Fe||Fe coin cell, asymmetrical Fe||Cu coin cell and EIS of the Fe||Fe symmetric cells show that the anode iron metal is stable with 1 M Fe(TOF)$_2$ as the electrolyte. Related discussion is shown in Supplementary Note 1 and Supplementary Note 2. In addition, although the theoretical voltage of the battery is 1.21 V, even when the battery is charged to 1.3 V, the oxidation peak at which divalent iron becomes trivalent iron ions does not appear. This is because the experimental Fe deposition voltage was −0.64 V in the aqueous electrolyte, which is −0.32 V lower than the HER thermodynamic potential[6]. For the Fe||C-PANI battery, there are two reduction peaks at 0.9 and 0.56 V, while Fe||PANI provides two reduction peaks at relatively lower voltages of 0.78 and 0.39 V (Fig. 2a). These can also be certified by the corresponding discharge plateaus (Fig. 2b). The C-PANI delivers a specific capacity of 209 mAh g$^{-1}$ at 5 A g$^{-1}$, which is higher than that of the conventional PANI (120 mAh g$^{-1}$ at 5 A g$^{-1}$) (Fig. 2b). The rate performance and cycling stability were also investigated to further reveal the potential of the Fe||C-PANI battery. As shown in Fig. 2c, d, the Fe||C-PANI battery exhibits good rate capabilities of 209–133 mAh g$^{-1}$ at specific currents ranging from 5 to 25 A g$^{-1}$; thus, even at the specific current of 25 A g$^{-1}$, a high capacity retention of 53.2% was achieved. When the specific currents was lowered from 25 A g$^{-1}$ to the initial 5 A g$^{-1}$, a capacity of 203 mAh g$^{-1}$ was still achieved. When the specific current, similar charge/discharge curves were obtained; in addition, a minor separation of the voltage plateau was observed, implying that Fe||C-PANI exhibits little electrode polarization.

Although the charge–discharge curves exhibit a triangle shaped voltage profile at high specific currents from 10 to 25 A g$^{-1}$, the CV curves of Fe||C-PANI show peaks and detectable charge/discharge plateaus at approximately 0.90 V and 0.4 V at a specific current of 5 A g$^{-1}$. The corresponding Coulombic efficiency (CE) of the Fe||C-PANI battery is 97% at 5 A g$^{-1}$ and 100% at 10–25 A g$^{-1}$, demonstrating high reversibility. In contrast, bare carbon cloth delivers a specific capacity of only 7 mAh g$^{-1}$ at the same specific current (Supplementary Fig. 11), suggesting that the carbon cloth substrate contributes a negligible capacity. The electrochemical capacity mainly originates from the C-PANI cathode. As shown in Fig. 2e, Fe||C-PANI exhibits good high-rate performance compared to that of state-of-the-art Fe-based aqueous batteries. Moreover, the Fe||C-PANI battery exhibits excellent cycling stability, which remained stable after 39,000 cycles with a remaining discharge capacity of 107 mAh g$^{-1}$ at a specific current of 25 A g$^{-1}$ with 84% capacity retention, and the corresponding charge–discharge profiles are shown in Supplementary Fig. 12; In contrast, the capacity of the Fe||PANI battery (Fig. 2f) decays quickly within 5,000 cycles due to electrochemical degradation of the conventional PANI. The Fe||C-PANI battery outperformed most reported aqueous metal batteries (Fig. 2g)[38–40]. Similar stability can also be observed with another sample at a loading mass of ~1.1 mg cm$^2$, as shown in Supplementary Fig. 13. The capacity fluctuation during 39,000 cycles is relevant to thermodynamic kinetics alteration in the electrolyte caused by temperature variation. To further confirm the high rate performance, batteries with high mass loadings (~3.5 mg cm$^2$ and ~8 mg cm$^2$) were tested, and the results are shown in Supplementary Figs. 14, 15.

## Charge-storage mechanism of C-PANI

Generally, aqueous batteries possess fast rate behaviour; the behaviour is attributed to proton storage, which exhibits fast conduction by the Grotthuss mechanism[41]. The proton movement is analogous to Newton's cradle, as local proton substitution occurs, leading to long-range transportation. A schematic diagram of the Grotthuss H$^+$ mechanism is shown in Supplementary Fig. 16. To investigate whether

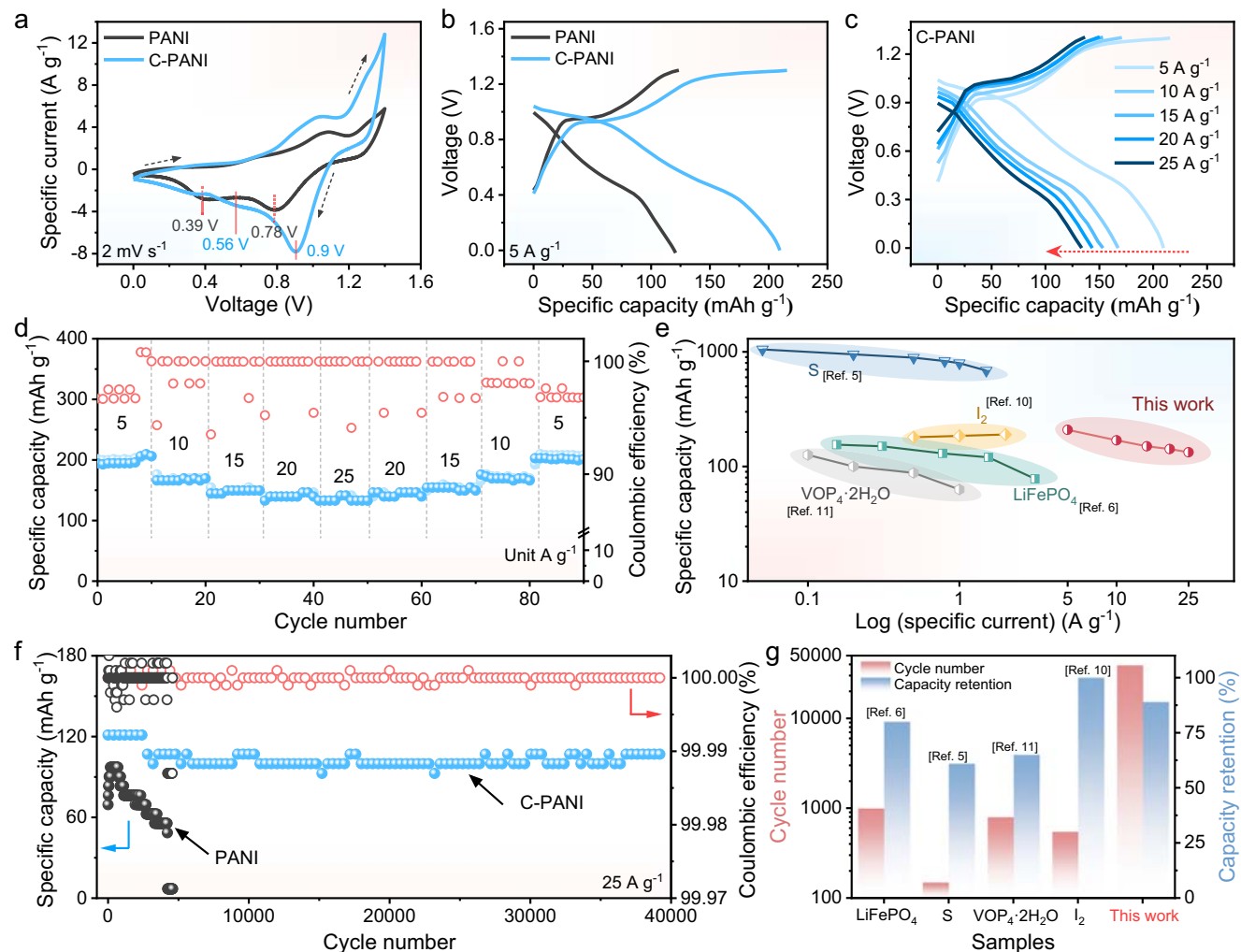

**Fig. 2 | The electrochemical performance of C-PANI. a** CV curves of Fe||PANI and Fe||C-PANI at 2 mV s⁻¹. **b** The discharging–charging profiles of PANI and C-PANI at a specific current of 5 A g⁻¹. **c** The discharging–charging profiles of C-PANI at specific currents ranging from 5 A g⁻¹ to 25 A g⁻¹. **d** The rate performance of C-PANI and corresponding Coulombic efficiency. **e** Specific capacity and specific current comparisons of Fe||C-PANI to previously reported aqueous Fe-ion batteries. **f** The long-cycle profiles for C-PANI and conventional PANI at 25 A g⁻¹. **g** Comparisons of cycle numbers and capacity retention of Fe||C-PANI with previously reported aqueous Fe-ion batteries. The test temperature is 28 °C ± 1 °C.

protons are involved in the reaction during the charging and discharging process of the C-PANI cathode, in-situ UV–Vis, in situ Raman, and ex situ XPS were performed. The series of UV–Vis absorbance curves correlated to various potentials are presented in Supplementary Fig. 17. When Fe||C-PANI was charged from 0.9 V to 1.3 V, two absorption bands at 430 nm and 830 nm, corresponding to the charged cationic species known as polarons, show a blueshift. When discharging from 0.9 V to 0 V, a redshift appears and extends to the wavelength at 980 nm, corresponding to the conversion of the C-PANI cathode from the fully oxidized state to the fully reduced state. The first and second discharge/charge curves of Fe||C-PANI are shown in Fig. 3a, and the corresponding in situ Raman shows that the characteristic band of quinoid segments at 1602 cm⁻¹ ($\nu_{C=C}$) undergoes a significant shift and progressively recovers to the initial state (Fig. 3b), demonstrating that the full discharge/charge reaction is highly reversible. The disappearance and emergence of existing peaks demonstrate that the C-PANI molecular structure has transformed. Specifically, the band intensity at 1476 cm⁻¹ (C = N) decreases and completely disappears during the first discharging process. When the first discharging and second discharging at 0.9 V, the band gradually reappears, and a shoulder band at 1488 cm⁻¹ emerges, indicating that the H⁺ (at 1476 cm⁻¹) and Fe (II) ions (1488 cm⁻¹) coordinate to the imine

nitrogen of the semioxidizedsemi-oxidized state (emeraldine state, EB) of C-PANI (Fig. 3c). The appearance of the shoulder band at 1488 cm⁻¹ is accompanied by a substitution reaction of iron ions in C-PANI by hydrogen ions (Fig. 3c), which appeared only after the first charge to 0.9 V, similar to the result of Ti covalently bonded to PANI[42]. The band at 517 cm⁻¹, assigned to out-of-plane C-N-C torsion, shows a similar change. During the first discharge process, the =N- in C-PANI is reduced to -N⁻-, which could interact with the ions containing Fe²⁺.

According to previous reports on H⁺ storage in organic positive electrode active materials in aqueous metal batteries[40,43,44], and the much smaller diameter of hydrated H⁺ than hydrated Fe²⁺, we speculate that H⁺ and Fe²⁺ coreact with C-PANI. This hypothesis is verified by the composition of the CV curves obtained for C-PANI in 0.1 M Fe(TOF)₂ (pH = 2.8), 0.1 M HTOF (pH = 0.5), and 0.1 M Fe(TOF)₂ at pH = 0.5 by adding HTOF electrolytes (Fig. 3d). The low concentration electrolyte of 0.1 M Fe(TOF)₂ was used to prevent further the production of protons due to Fe(TOF)₂ hydrolysis, which would affect the test results. In 0.1 M Fe(TOF)₂ electrolyte, two reduction peaks at approximately 0.15/−0.2 V vs. Ag/AgCl and two oxidation peaks at 0.46/1.0 V vs. Ag/AgCl were observed during the sweep process, corresponding to the conversion of the fully reduced state (leucoemeraldine, LE) to the fully-oxidized state (pernigraniline salt, PNS) of

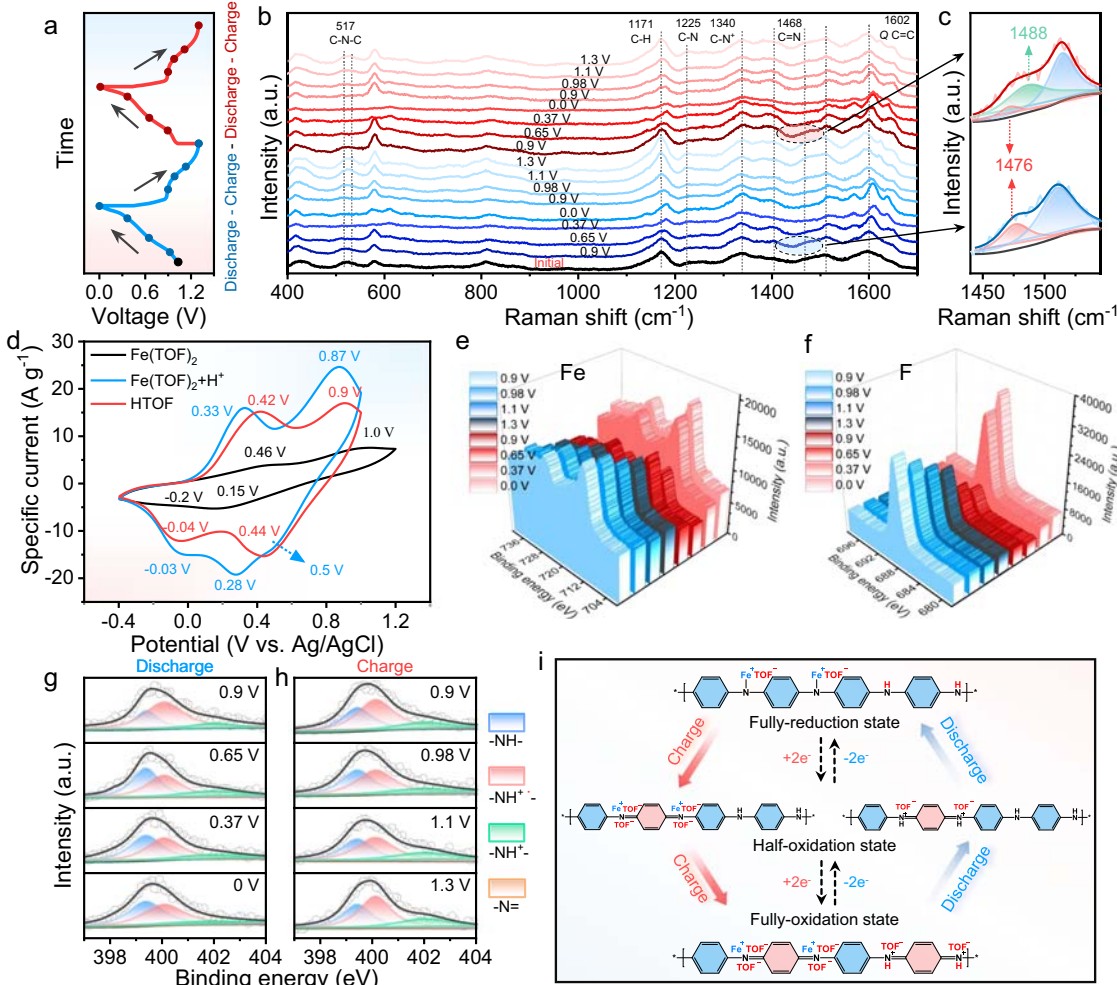

**Fig. 3 | Investigation on the C-PANI charge storage mechanism. a** The first and second discharge/charge curves of Fe‖C-PANI batteries using 1 M Fe(TOF)$_2$ as the electrolyte. **b** In situ Raman spectra of the C-PANI cathode were obtained at different discharging/charging states during the first two cycles. **c** Raman spectra of the first discharge at 0.9 V (down) and the second discharge at 0.9 V (top). **d** CV of the same C-PANI electrode in 0.1 M Fe(TOF)$_2$ (pH = 2.8), 0.1 M HTOF (pH = 0.5) and 0.1 M Fe(TOF)$_2$ at pH = 0.5 by adding HTOF electrolytes in a three-electrode cell at 50 mV s$^{-1}$ with platinum as the counter electrode and Ag/AgCl as the reference electrode. Ex situ XPS spectra of (**e**) Fe 2$p$ and (**f**) F 1$s$ of C-PANI cathodes obtained at charging/discharging state. The intensity of the Fe 2$p$ peak and F 1$s$ peak is corrected by the normalization of the intensity of N 1$s$. Ex situ XPS spectra of N 1$s$ of C-PANI cathodes obtained at the state of (**g**) discharging and (**h**) charging. (**i**) Schematic diagram of the charge/discharge mechanism of Fe‖C-PANI.

C-PANI. However, in 0.1 M HTOF electrolyte, the two reduction peaks shift up to 0.44/−0.04 V vs. Ag/AgCl, and the oxidation peaks shift to 0.42/0.9 V vs. Ag/AgCl. Of note, three subsequent reduction peaks appear at approximately 0.5, 0.28, and −0.03 V vs. Ag/AgCl in 0.1 M Fe(TOF)$_2$ at pH = 0.5 with the addition of HTOF electrolytes, which implies that three reactions occur during the discharge progress. It can be inferred that the peaks at 0.5 V in 0.1 M Fe(TOF)$_2$ at pH = 0.5 and at 0.44 V in HTOF correspond to the reaction between H$^+$ and C-PANI. Additionally, only two oxidation peaks were observed at 0.33 V and 0.87 V in 0.1 M Fe(TOF)$_2$ at pH = 0.5; this occurred because the H$^+$ reaction oxidation and ions containing Fe$^{2+}$ reaction oxidation overlap and because the steps occur too fast to distinguish. These results certify that H$^+$ and ions containing Fe$^{2+}$ coparticipate in the charge/discharge reaction of the C-PANI cathode. According to the above analysis, the fast rate behaviour of the Fe‖C-PANI aqueous cells is attributed to the high conductivity of C-PANI and the Grotthuss H$^+$ transport mechanism.

The ex situ XPS revealed that the intensity of iron decreases gradually as the potential increases from 0.9 V to 1.3 V during charging. Interestingly, during discharging, the intensity of the iron at 0.9 V was almost identical to that of 1.3 V, while the intensity dropped to a minimum at 0.65 V, and gradually increased to the maximum from 0.65 V to 0.0 V. In other words, in the charging process (C-PANI from the full reduction state to the oxidation state) and discharge to the semioxidized state, the iron ion is gradually removed from C-PANI until the iron ion content reaches the lowest level in the semioxidized state (Fig. 3e). Peculiarly, as C-PANI is converted from the oxidized state to the semioxidized state, iron ions in C-PANI are gradually exchanged by hydrogen ions simultaneously, which leads to the lowest iron ion content in semioxidized C-PANI at 0.65 V. In the following process of discharge from the semioxidized state to the fully reduced state, iron ions are reverse embedded in the C-PANI and react with it. The intensity of F 1$s$ exhibits the same tendency as Fe 2$p$ during charge and discharge (Fig. 3f). A more reasonable explanation could be that the monovalent complex cation formed by Fe$^{2+}$ and TOF$^-$ at a 1:1 ratio, interacts with C-PANI in charge and discharge, which can mitigate the strong charge densities of divalent Fe$^{2+}$ and reduce the strong electrostatic interaction with the host materials, leading to fast reaction kinetics and enhanced electrochemical performance[40]. To further confirm this observation, we obtained Raman measurements for the Fe(TOF)$_2$ electrolytes with various concentrations and Fe(TOF)$_2$ powder. This was performed to clarify the possible Fe$^{2+}$ dissolution

structure in the 1.0 M $Fe(TOF)_2$ electrolyte, which affects the charging and discharging process (Supplementary Fig. 18). As shown in Supplementary Fig. 19, as the $Fe(TOF)_2$ concentration increased from 0.5 M to 1 M, the strong hydrogen bond at 3555 $cm^{-1}$ becomes more intensive, indicating that more $TOF^-$ participates in the coordination shell of the cation[37]. Additionally, the Raman peak at 1034 $cm^{-1}$, belonging to the vibration of the O=S bond of $TOF^-$, can be deconvoluted to free $OTF^-$ (O=S signal at 1032 $cm^{-1}$) and contact ion pairs (CIP, Fe–O=S signal at 1038 $cm^{-1}$)[45,46]. Compared to the 0.5 M $Fe(TOF)_2$ electrolyte, more Fe–O=S appeared in the 1 M Fe $(TOF)_2$, also indicating that more $OTF^-$ anions participate in the solvated structure of $Fe^{2+}$ in the 1 M $Fe(TOF)_2$ electrolyte (Supplementary Fig. 20). The occurrence of the $TOF^-$ containing solvated structure increases the possibility of the $Fe(TOF)^+$ complex ions coparticipate in the reaction with C-PANI during the discharge process. To further confirm this hypothesis, the C-PANI cathode material was characterized in the fully charged state and fully discharged state by FT-IR spectroscopy (Supplementary Fig. 21). A very broad S=O vibrational peak appears near 1030 $cm^{-1}$ in the discharge state, suggesting that the chemical environment of S=O adsorbed on the C-PANI surface changes in the fully discharged state. The broad peak contains both $Fe(TOF)_2$ adsorbed on the surface and $Fe(TOF)^+$ reacted with C-PANI. To prevent physical adsorption of $Fe(TOF)_2$ on the C-PANI surface from disrupting the test results, the samples were polished before the measurement. After polishing the C-PANI electrode, a shoulder peak appears at approximately 1022 $cm^{-1}$ due to alteration of the S=O chemical environment, unravelling the couptake of $TOF^-$ and $Fe^{2+}$ and the reaction with C-PANI. In the following fully charged state, the S=O peak is present at 1030 $cm^{-1}$ instead of 1021 $cm^{-1}$, suggesting that the broad 1030 $cm^{-1}$ is from $Fe(TOF)_2$ adsorbed on the surface.

Moreover, the participation of $Fe(TOF)^+$ in the reaction was further analysed by XPS. The high resolution Fe $2p$ XPS spectra of the fully discharged C-PANI show five peaks at approximately 712.3 eV, 715.1 eV ($Fe^{2+}$), 720 eV (satellite), 726 eV, and 730.1 eV ($Fe^{2+}$), correlating with the adsorption of $Fe(TOF)_2$ on the surface of C-PANI (Supplementary Fig. 22). After 2 min of etching, the two new peaks at 710.0 eV and 724.2 eV emerged and intensified as the etching time was extended to 4 min, implying that $Fe^{2+}$ coordinates with the N of C-PANI and forms an Fe-N structure after being fully discharged[47]. Furthermore, the high resolution S $2p$ spectra also showed a variation after discharge. As shown in Supplementary Fig. 23, three peaks at 169 eV, 167.5 eV and 162.5 eV, which were not observed for C-PANI in fully discharged state, appeared and intensified with increasing etching time. This is due to the formation of coordination between $Fe^{2+}$ and $TOF^-$ to form $Fe(TOF)^+$ complex ions, which react with N in the discharged state, leading to a change in the S chemical environment. The coordination mode is also shown in the ball-and-stick structure (inset of Supplementary Fig. 23).

The reversible variations in C-PANI during charging/discharging can be confirmed by ex situ XPS spectra of N 1$s$ of C-PANI cathodes during the reversible process (Fig. 3g, h). During the discharging process (from up to down) shown in Fig. 3g, the intensity of the protonated amine (–NH$^{+}$–, 400.3 eV) and protonated imine (–NH$^{+}$–, 402.4 eV) decreases, and the intensity of the benzenoid amine (–NH–) increases, indicating C-PANI conversion from the oxidation state to the reduction state. During the following charging process (from up to down) shown in Fig. 3h, the intensity of protonated amine (–NH$^{+}$–, 400.3 eV) and protonated imine (–NH$^{+}$–, 402.4 eV) increases, and the intensity of benzenoid amine (–NH–) decreases.

The proposed charging and discharging mechanism is shown in Fig. 3i. To verify whether $Fe^{3+}$ reacts with C-PANI in charging and discharging processes, C-PANI cyclic voltammetry curves were recorded in 0.1 M $Fe(TOF)_3$ electrolyte (containing $Fe^{3+}$ cations) (Supplementary Fig. 24). The peak potentials are nearly identical for the pairs 0.13 V/ 0.9 V and −0.2 V/0.39 V pertaining to $Fe^{2+}$ participation in the conversion reaction in C-PANI. However, for $Fe(TOF)_2$, the gaps of peak

potentials within each pair are small. The variation in values is due to the different pH values in the electrolyte[48]. The standard electrode potential of $Fe^{3+}/Fe^{2+}$ is +0.77 vs. the standard hydrogen electrode. $Fe^{3+}$ ions were reduced to $Fe^{2+}$ ions before the first reduction peak appeared (at 0.39 V)[49].

## Electrochromic battery performance analysis

To date, numerous aqueous electrochromic batteries have focused on optical transmittance and developed these batteries for smart windows[50] and information displays[42]. Moreover, flexible reflective electrochromic devices are widely used in aerospace thermal control and camouflage[51,52]. These results encouraged us to develop dual-function flexible reflective electrochromic batteries (FREBs) that can simultaneously store energy and perform electrochromic functions. The preparation route of the FREB is illustrated in Fig. 4. The flexible conductive Au porous membrane (FCAPF) was prepared by evaporating pure gold on the nylon 66 microporous membrane; then the C-PANI cathode was chemically oxidized on the Au nylon 66 membrane (Supplementary Fig. 25). Gold was selected because it exhibits the following properties: 1. High electrochemical stability; 2. High electrical conductivity; 3. High IR reflectivity. The infrared reflectance spectra of Au-coated nylon66 are shown Supplementary Fig. 26. The flexible anode was fabricated by electrodeposition of metal iron on a carbon cloth substrate. SEM images and XRD confirmed that the deposition was successful (Supplementary Figs. 27, 28). Then, the polyacrylamide hydrogel (Supplementary Fig. 29) and non-woven composite film electrolyte containing 1 M $Fe(TOF)_2$ (the preparation process is provided in the Methods section) were sandwiched between electrodeposited iron and C-PANI electrodes to assemble a FREB, as shown in Fig. 4, and the detailed preparation process is reported in the Methods section.

The CV analysis of the FREB revealed the reaction potential in Fig. 5a. An irreversible oxidation peak appeared above 1.3 V, which is due to electrochemical peroxidation and degradation of C-PANI with the hydrogel as the electrolyte[18,53]. The discharge capacity of 163 mAh $g^{-1}$ is delivered at 5 A $g^{-1}$ of the FREB device (Fig. 5b, c). With gradually increasing current rate, the FREB electrode exhibits an average reversible discharge capacity of 136, 126, 114, and 107 mAh $g^{-1}$ at 10, 15, 20, and 25 A $g^{-1}$, respectively. The discharge capacities recovered fully when the current decreased from 25 A $g^{-1}$ to the initial 5 A $g^{-1}$, revealing a good rate capability. Figure 5c shows the galvanostatic discharge/charge profiles of the FREB device at various specific currents. Notably, at a high current rate of 25 A $g^{-1}$, the plateau was well discerned in both charge/discharge processes, and a high specific capacity of 107 mAh $g^{-1}$ was obtained (Fig. 5c). After 27,000 cycles, the FREB maintained 82% of its maximum capacity, demonstrating an excellent long-term cycling performance (Fig. 5d). The coulombic efficiency fluctuated after 27,000 cycles, which may be caused by the following reasons. 1. The flexible electrochromic batteries are not 100% sealed as coin-type cells, and after 20000 cycles, the hydrogel electrolyte gradually loses its water content; 2. The electrochemical degradation of C-PANI[18,53]. In comparison with other reported flexible aqueous hydrogel batteries and electrochromic batteries, the FREB presents excellent long-term and stable cycling performances (Supplementary Table 1).

The visible-region electrochromic performance of the FREB was characterized by reflectance spectra. Figure 5e shows the visible and near-infrared region reflectance spectra of the FRED in the charge/ discharge state. The device in charged state exhibited a maximum reflectance in the region of 400–1300 nm, which is assigned to the C-PANI colour, and blueshifted by 80–160 nm compared with the corresponding spectra of the discharged state; this shift is attributed to the mixed spectra of the gold electrode and the reduced C-PANI transparent state[54]. Furthermore, this band is broader than that in the discharged state, suggesting that quinoid fragments formed. Figure 5f shows the CIE 1931 xy chromaticity coordinate changes at the

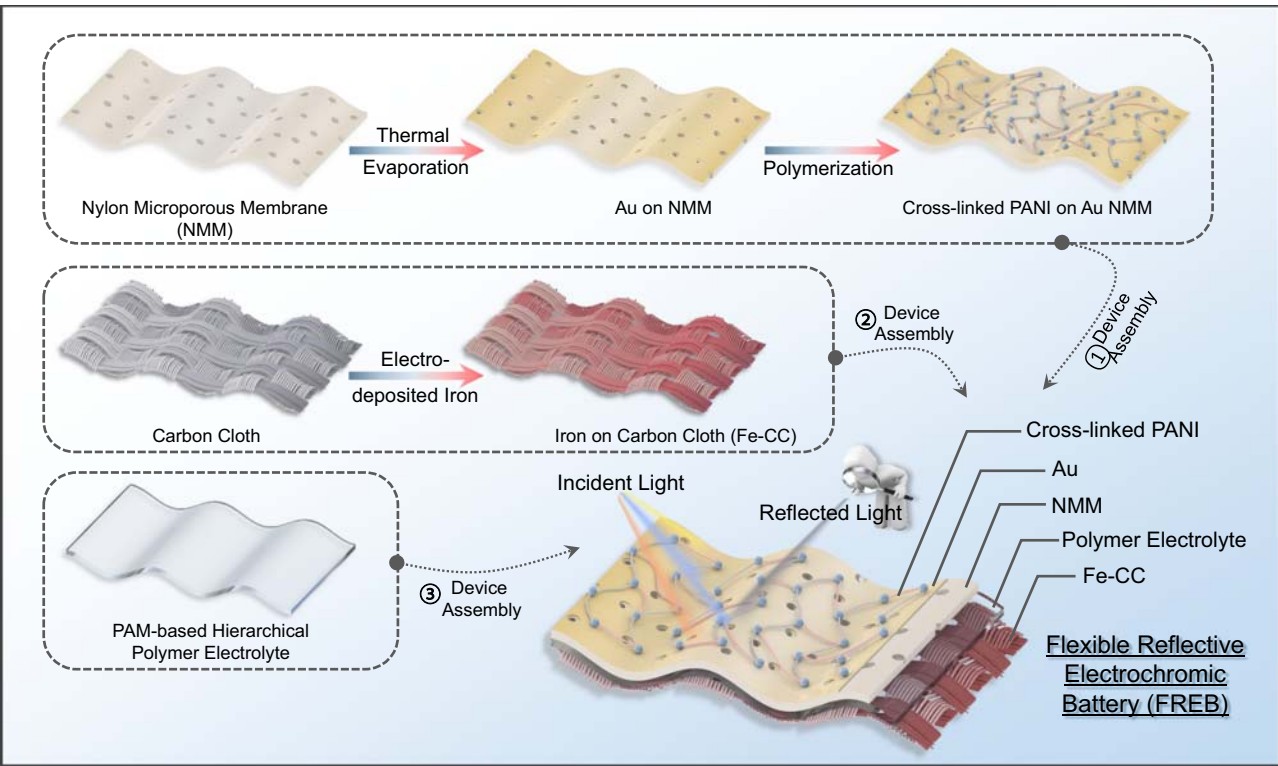

**Fig. 4 | Schematic of the preparation route for flexible reflective electrochromic batteries.** Incident light is partly absorbed and partly reflected when it hits the electrochromic device. The reflected light is the part of the light that is reflected by the flexible reflective electrochromic batteries.

charge/discharge state as calculated from the in situ spectra of Fig. 5e, demonstrating that the CIE colour space of FREB tuneable from charge to discharge states. The device exhibited electrochromic behaviour due to C-PANI,and its colour changed from yellow in the discharge state to green in the charge state (Fig. 5g). To further evaluate the IR regulation performance, in situ spectral emissivity characterizations were performed from 0 V to 0.9 V. The contour map of emissivity ($\varepsilon$) evolution in the wavelength range of 2.5–25 μm is shown in Fig. 5h. $\varepsilon$ gradually increases in the wavelength range of 2.5–7.5 μm with increasing voltage, which proves that the FREB possesses the ability to dynamic regulate emissivity in the IR region.

To demonstrate the thermal management performance of the FREB, emissive power curves (in 2.5–25 μm) of the FREB were obtained with a decreaseing voltage from 0.9 V to 0 V, as shown in Fig. 5i. The corresponding ideal black body spectra (298.15 K) are presented in Supplementary Fig. 30. Emissive power of the FREB is 4.5 W/μm m² at 4.5 μm increased to 12.1 W/μm m², and the 3D contour map exhibits the same trend (Fig. 5j), suggesting promising application potential. The detailed calculation method is reported in the Methods section. As the battery is a reflective electrochromic battery, it exhibits the following additional advantages: (1) The device may show different colours, so it can blend into the surrounding environment; (2) The batter exhibits a capacity of thermal management; (3) The energy storage state can also be expediently monitored by colour exhibited by the C-PANI.

## Discussion

In summary, C-PANI was successfully prepared by a chemical oxidation process. Compared to conventional PANI that is electrochemically degradable, the C-PANI exhibits improved stable complex ions Fe(TOF)$^+$ and H$^+$ costorage performance, delivering an excellent long cyclic life (remain 107 mAh g$^{-1}$ after 39,000 cycles at 25 A g$^{-1}$) and high rate performance (120 mAh g$^{-1}$ at 25 A g$^{-1}$). The high electrochemical performance can be attributed to the following reasons: first, the crosslinker improves the electronic conductivity of C-PANI and

realizes fast Fe(TOF)$^+$ accessibility; second, proton diffusion is fast through the Grotthuss mechanism. Third, the Fe$^{2+}$ ions are bonded to CH$_3$SO$_3^-$ to form mixed-cation Fe(TOF)$^+$, which can mitigate the strong charge densities of divalent Fe$^{2+}$ and reduce the strong electrostatic interaction with host materials, leading to fast reaction kinetics and enhanced electrochemical performances. Additionally, as the battery is a reflective electrochromic device, it can monitor the energy storage state and thermal management.

## Methods
### Materials
All solvents and chemicals were of analytical grade and used without further purification. Aniline was distilled under reduced pressure and stored under nitrogen. Diphenylamine (99%, Innochem), melamine(99%, Innochem), iron(II) trifluoromethanesulfonate (Fe(CF$_3$ SO$_3$)$_2$, 97%, Innochem), iron(II) perchlorate hydrate (98%, Sigma-aldrich) Fe foil (99.99%, Qinghe County Dingyuan Metal Products Co., Ltd.), ammonium persulfate(99.99%) hydrogel and nonwoven composite film electrolyte were used.

### Synthesis of the crosslinked PANI electrode and conventional PANI electrode
Cross-linked PANI was synthesized by chemical oxidative polymerization. First, 0.3 g aniline and 9 mg melamine were dissolved in 100 mL of 1.0 M HCl solution, 21.8 mg diphenylamine was dissolved in 2 ml ethanol. The diphenylamine solution was added to the solution containing aniline and melamine and then placed in the incubator until the temperature was approximately 0–5 °C. Finally, the solution containing 0.75 g ammonium persulfate, used as the oxidizing agent, was addeddropwise into the above solution, and the solution was stirred at 0 °C for 12 h.

Carbon cloth or pure gold on the nylon 66 microporous membrane was put into the solution to prepare electrodes. The C-PANI electrodes obtained were washed repeatedly with deionized water and

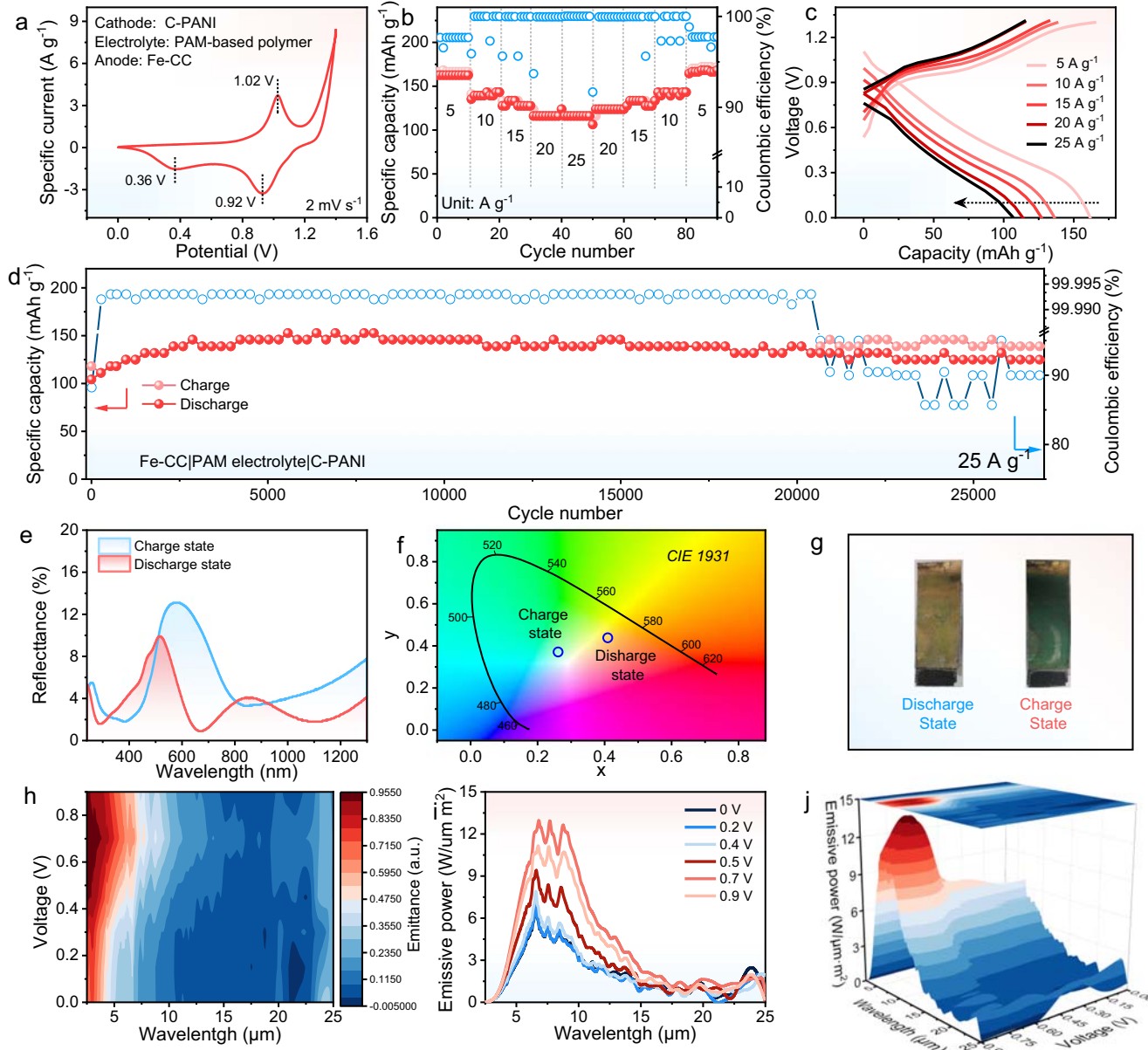

**Fig. 5 | Electrochemical performance of FREB. a** CV curves of FREB at 2 mV s⁻¹. **b** The rate capability of the FREB at various rates. **c** Corresponding galvanostatic charge/discharge profiles at various rates. **d** Long-term cycling performance and the corresponding coulombic efficiency at 25 A g⁻¹. **e** Reflectance spectrum, and (**f**) CIE 1931 xy chromaticity coordinate changes at the charge/discharge state. **g** Optical photo of FREB at various states of charge. **h** In situ spectral emissivity contour map, (**i**) emissive power and (**j**) emissive power contour map of the FREB at the corresponding voltages from 2.5 μm to 25 μm. The electrochemical test temperature is 25 °C ± 1 °C.

ethanol, and the obtained solid was dried at 60 °C for 10 h in a vacuum oven. For comparison, PANI without melamine and diphenylamine was synthesized under the same reaction conditions. The active mass loading of C-PANI on the carbon cloth was 1.02–1.1 mg/cm².

**Preparation of hydrogel and non-woven composite film**

In a typical run, 2 g acrylamide monomer powder was added to 20 g deionized water and stirred for 30 min at 25 °C to fully dissolve the powder. Then, 10 mg of potassium persulfate (initiator) and 2 mg of N,N'-methylenebis(acrylamide) (cross-linker) were added into the above solution and stirred for 1 h at 25 °C. After that, the mixture was stirred at 40 °C for 2 h, evacuated, injected into a nonwoven film and heated at 60 °C for 2–3 h. Then, a crosslinked 3D framework structure filled by a polyacrylamide (PAM)-based electrolyte was formed in the nonwoven pores. Finally, the as-prepared polymer film was soaked in

an aqueous solution of 1 mol L⁻¹ Fe(SO₃CF₃)₂ for 12 h in a nitrogen-filled glove box to achieve the saturation.

**Preparation flexible reflective electrochromic batteries**

A flexible iron electrode was prepared by electrodepositing iron onto carbon cloth (CC) in a 1 M solution of Fe(ClO₄)₂ in propylene carbonate (99%, Alfa). The electrodeposition was conducted by cyclic voltammetry (CV) with a potential range of −0.7 to 0.3 V and a sweep rate of 10 mV s⁻¹ for 30 cycles in a three-electrode system in a nitrogen-filled glovebox; the CC served as the working electrode, and a platinum (Pt) sheet and Ag/AgCl were used as the counter electrode and reference electrode, respectively. Then, the PAM-based electrolyte was sandwiched between flexible iron and the back side of the C-PANI electrodes to fabricate flexible reflective electrochromic batteries.

## Electrochemical measurements

All electrochemical measurements were conducted at an average temperature of 28 °C ± 1 °C. The battery performances were measured in coin-type cells, and dual-function flexible reflective electrochromic batteries were measured in a flexible configuration. The electrochemical energy storage tests were carried out in a nitrogen-filled glovebox (MIKROUNA shanghai Company Ltd). In an air environment, the 0.3 mm Fe foil was first polished with sandpaper (600#) to remove the mineral oil on the surface that prevents the iron from being oxidized. To prepare the $Fe(CF_3SO_3)_2$ electrolyte, deionized water was boiled to eliminate dissolved $O_2$. The degassed water was transferred to a nitrogen-filled glovebox (water content: 0.9 ppm, and oxygen content: 1.2 ppm) to prepare the 1 M $Fe(CF_3SO_3)_2$ electrolyte. The charge/discharge performances of the cells were measured with a Neware battery testing system (Shenzhen China), and Long-term cycling data results were analysised by Microsoft Excel and Origin and using skip points (100 data points). Cyclic voltammetry was performed on a CHI760E electrochemical work station. The Fe||Fe symmetric coin cells were cycled and kept at open circuit for 2 h prior to the (electrochemical impedance spectroscopy) EIS measurement at the open-circuit voltage. The EIS spectra were obained under potentiostatic mode within the frequency range of $10^{-2}$–$10^5$ Hz (85 data points). The amplitude of the applied current oscillation is 5 mV. The electrical conductivity measurement of C-PANI was carried out with a four-probe tester (RTS-4) from 4 Probes Tech Ltd. The samples containing 0.3 g PANI or C-PANI powder were prepared in the infrared tablet press and tested at a current range of 10 μA in the four-probe test system. Galvanostatic discharge–charge profiles were performed by Neware CT-4008T battery measurement system.

## Physicochemical characterizations

The samples were characterized by SEM (Hitachi S-4800, 5 kV), XPS (Kratos Analytical AXIS-Ultra with monochromatic Al Kα X-ray) and TEM (JEM-F200). C-PANI and PANI were polymerized directly onto the surface of carbon cloth as a working electrode. Samples for ex situ XPS measurements at different voltages were prepared in a nitrogen-filled glovebox, kept dry and removed for testing without the need for transport procedures. For in situ Raman testing, the Fe||C-PANI cell was assembled inside a nitrogen-filled glovebox using an in situ Raman spectroscopy electrochemical cell (C031-1, GaossUnion). All binding energy values of the XPS results were referenced to the C 1s peak of carbon at 284.8 eV. Raman spectroscopy (HORIBA iHR550, excitation light of ~633 nm) and FTIR spectra (Bruker Hyperion FTIR spectrometer). The spectral emittance of the devices was tested by an INVENIO-R (Bruker) FTIR spectrometer in the spectral range of 2.5–25 μm (equipped with an integrating sphere). The C-PANI for tTEM was scraped off the carbon cloth with a knife.

**Emissivity calculation.** The radiated thermal energy per unit area from a hot surface is characterized by the Stefan−Boltzmann law, $P = \varepsilon \sigma T^4$ where ε is the emissivity of the surface, σ is the Stefan−Boltzmann constant, and T is the temperature of the surface. The emissivity is the only material-dependent parameter that varies with the wavelength and temperature. At thermodynamic equilibrium, Kirchhoff's radiation law connects the wavelength-specific thermal emissivity with the optical absorption of the surface as $\varepsilon(T,\lambda) = \alpha(T,\lambda)$. According to Planck's law, emissivity is a nondimensional parameter for quantifying thermal radiation, and the value of ε can be calculated by weighting $(1-R(\lambda))$ (namely spectral emittance) with the black body spectrum according to the following two Eqs. (1) and (2)[55,56]

$$B(\lambda) = \frac{c_1 \lambda^{-5}}{\exp\left[c_2/(\lambda T)\right] - 1} \tag{1}$$

$$\varepsilon = \frac{\int_{\lambda_{min}}^{\lambda_{max}} (1 - R(\lambda))B(\lambda)d(\lambda)}{\int_{\lambda_{min}}^{\lambda_{max}} B(\lambda)d(\lambda)} \tag{2}$$

where $c_1$ is the first radiation constant ($3.7418 \times 10^8$ W μm⁴ m⁻²), $c_2$ is the second radiation constant ($1.4388 \times 10^4$ μm K), λ is the wavelength, and T is the temperature.

Conducting polymer normal reflectance can be expressed according to the Drude free electron theory and the Hagen−Rubens approximation at a low frequency as follows[57]:

$$R(\lambda) = 1 - (2\varepsilon/\pi\sigma)^{1/2} \tag{3}$$

where $R(\omega)$ is the normal reflectance at the angular frequency ω, and σ is the electronic conductivity.

According to Kirchhoff's law of thermal radiation, a material's frequency-dependent emittance is equal to its frequency-dependent absorbance in thermal equilibrium. In the entire measured wavelength range, ε can be defined as

$$\varepsilon = 1 - R \tag{4}$$

In situ visible electrochromic measurements were performed using an experimental setup that was produced in-house in combination with a CHI 760D electrochemical workstation (Shanghai Chen hua Instrument Co. Ltd.). The experimental setup was sealed during testing. The UV–Vis spectra were tested (HATACHI UH4150). C-PANI and Fe were used as the cathode and anode, respectively. 1 M $Fe(CF_3SO_3)_2$ were used as electrolytes (Supplementary Fig. 31), and the CV measurements were performed at an average temperature of 28 °C ± 1 °C between 0 and 1.3 V at various scan rates in a climatic/environmental chamber.

## Data availability

The data that support the findings of this study are available within the article (and its Supplementary Information files) and from the corresponding authors upon reasonable request.

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

## Acknowledgements

This research was supported by Guangdong Basic and Applied Basic Research Foundation (grant number: 2020A1515110896, 2020A1515110442, 2019A1515110980), National Natural Science Foundation of China (22005207), the Open Research Fund of Songshan Lake Materials Laboratory (2021SLABFN04), National Key R&D Program of China (2019YFC1905801).

## Author contributions

C.Z. led the team and conceived the concepts for the research project. H. Li., C.H., supervised the project. H. Lv. designed and performed the experiments for the synthesis of C-PANI, and the characterization including FT-IR, Raman, XPS, and the mechanism study. X.Y. aided in the characterization and analysis of UV–vis–NIR. Z.W. aided in the electrocromic experiments and analysis. Y.Z. and Z.T. contributed to the scientific discussions and provided technical support. All authors discussed the results and comments on the manuscript. H. Lv. wrote the paper.

## Competing interests

The authors declare no competing interests.
