## [Peer Review File · Nature Communications]

REVIEWER COMMENTS

Reviewer #1 (Remarks to the Author):

1. Using PANI as cathodes in proton-based energy-storage systems is a routine paradigm example. This has been fully confirmed by many research cases in aqueous battery and faradaic pseudocapacitors. Thereby, the concept of metal anode//organic cathode battery should not be highlighted.
2. The proposed co-doping of Fe(TOF)⁺ into PANI is not convincing, as reflected by Fig. 3(d). For Fe-free electrolytes, there are still distinct CV profiles present; while in Fe(TOF)⁺ cases, readers still notice weak but very similar current signals. The current response proves the cathode performance is highly correlated with the proton concentration rather than ferruginous ion species.
3. The in-situ Raman detections (Fig. 3(b)) neither show any direct evidences. Though ex-situ XPS spectra reveal the chemical state change of Fe on PANI surfaces, the bulk insertion of Fe(TOF)⁺ is still unknown.
4. Authors neglect significant research on the Fe anode side. Though using proton-rich solution (pH: ~0.5) benefits the redox reactions of PANI, the use of highly acidic electrolyte is quite detrimental to Fe anodes, leading to parasitic reactions like severe metallic corrosion, the build-up of thick interfacial passivation layers, etc.
5. The super-long 39,000-cycle battery lifespan (coulombic efficiency always approaches 100%; Fig. 2f) is surprising but hardly understood, since the Fe electroplating/stripping is not an ideally reversible process (actually, there are inevitable side reactions occurring, particularly at the anode side). So, how to explain this result?
6. Likewise, the superb rate capability (120 mAh g⁻¹ retained under 25 A g⁻¹) is also confusing. How can authors account for such outstanding rate performance of battery devices built by conducting polymers with less electronic conductivity than carbons? High mass loading is required to demonstrate the high rate performance.
7. The charge/discharge voltage profiles (Fig.2b) are inconsistent to CV plots (Fig.2a). The potential plateaus become slope-like profiles. Note the voltage profiles are seemingly close to a triangle shape, as often recorded in pseudocapacitor testing. Also, a huge voltage polarization is noticed evidently.

Based on above concerns, this reviewer can hardly recommend its publishing on Nature Communications.

Reviewer #2 (Remarks to the Author):

The authors prepared highly conductive cross-linked PANI (C-PANI) using melamine as the crosslinker and applied them as electrodes for aqueous iron batteries. The Fe//C-PANI batteries exhibited fast reaction kinetics and record-breaking battery performance including super-high rate capability (120 mAh g⁻¹ at 25 A g⁻¹) and a super-long 39000 cycle life, which represents a solid advance on aqueous Fe batteries. More importantly, the proton-Fe(TOF)⁺ co-storage mechanism was demonstrated. Multifunctional Fe//C-PANI electrochromic batteries were also developed to integrate both electrochromism in the visible and middle-Infrared and energy storage, demonstrating great potential in next-generation smart battery technologies. The experimental results are convincing and are analyzed in sufficient depth. Therefore, I recommend that this work can be published in Nature Communications. And some revisions are needed to improve the manuscript further before the acceptance.

1. Why choosing 1 M Fe(TOF)₂ as the electrolyte? FeSO₄ is a commonly used electrolyte, and it can be much cheaper. Since one of the merits of Fe batteries is its low cost, the authors should explain their choice on electrolyte more profoundly.
2. The standard electrode potentials of the Fe³⁺/Fe²⁺ and Fe²⁺/Fe couples are +0.77 and -0.44 V versus SHE, respectively, which gives rise to a theoretical electrochemical window of ≈1.21 V for a Fe metal battery. However, why did the author charge the battery to 1.3 V? Is there any problem with that?
3. In Fig. 3d, are the CV results for the same electrode sample? If yes, please clarify; if not, please express the vertical axis units in terms of current density.
4. Why gold is coated on the surface of nylon 66? Is it okay if it is another metal? Please further clarify this point.
5. The forward and backward peaks in CV curves should be well assigned for better clarity (Figure 5a), the reactions should be well marked.
6. Why are there some rise and fall in the long cycle test in Fig. 2f?
7. For the TEM observation of C-PANI, how to peel it off from the substrate?

Reviewer #3 (Remarks to the Author):

In this article, the author developed an aqueous iron battery with long cycle life and high rate capability. Given its innovative and excellent electrochemical properties, I think it could be published in Nature communications. Aqueous iron-ion batteries are in the early stage of development, there is a lack of high-performance Fe-bearing cathode materials coupled to ferrous metal anodes. The battery of iron metal has poor cycle stability and low Coulombic efficiency. The C-PANI cathode used in this paper has not been explored in previous iron batteries, which is relatively innovative. The developed battery has an extraordinary cycle life of 39,000 cycles, high rate capability, and an initial specific capacity of 209mAh/g at 5 A/g, with excellent electrochemical performance. But there is still room for improvement in this paper, details as follows.

1. Isn't the vibration of the -NH= structure at 1139cm^{-1} in Fig. 1b?
2. Figure 2a can be more standardized, and the position and scanning direction of the redox peak can be marked.
3. Page 9, line 14, 1476 cm^{-1} (C=N) The shoulder strap at 1488 cm^{-1} appeared only after the first charge to 0.9V.
4. What is the irreversible peak at 1.3 V in Fig. 5a assigned for?
5. Figure 5d, it can be seen from the figure that the Coulomb efficiency of the battery has decreased significantly after 20,000 cycles. What is the specific reason? Why the Coulomb efficiency value fluctuated with fixed step?
6. The detailed Fe storage mechanism should be investigated. XPS spectra of Fe should be included for Fe-stored PANI.

Manuscript reference number: NCOMMS-22-24526A

In this Response Letter, we have addressed and clarified all the helpful and valuable comments from referees. With a substantial amount of new results added to the revised manuscript according to the referees' comments, we are very grateful to the reviewers for their dedication to this work.

Below, we provide our point-by-point responses to the referees' comments, where the original comments are shown in black, and our responses are shown in blue.

Reviewer #1 (Remarks to the Author):

1. Using PANI as cathodes in proton-based energy-storage systems is a routine paradigm example. This has been fully confirmed by many research cases in aqueous battery and faradaic pseudocapacitors. Thereby, the concept of metal anode//organic cathode battery should not be highlighted.

Reply: Thanks very much for your comments. We agree with the reviewer that the concept of metal anode/organic cathode is not the core of our paper and should not be highlighted. Our modification to polyaniline has led to significantly improved cycling stability. Following the reviewer's suggestion, we have revised the title to "*Aqueous iron batteries with 39 000-cycle lifespan and superior rate capability*" in our revised manuscript.

Though it has been reported in other energy-storage devices, polyaniline has never been explored as cathode material for aqueous metal-iron batteries, and our cross-linking strategy results in the super stability of the iron batteries. In addition, the related energy storage mechanism has also not been unveiled for iron batteries. Aqueous iron-ion batteries are in the early stage of development, the development and application were limited due to the lack of proper cathode materials coupled with Fe metal anodes.

The main message of this paper is to report a kind of cross-linked polyaniline that has excellent battery performance in aqueous iron batteries. The electrochemical stability of the polyaniline is significantly increased by a strategy of introducing cross-linked sites in the polyaniline. In contrast, polyaniline without effective cross-linking exhibits poor stability as a cathode material for aqueous iron batteries.

Considering the difficulties to develop cathode materials for iron batteries and our remarkable performance achievements, we believed this manuscript represents remarkable progress in iron batteries.

We have revised the title in our revised manuscript and added related discussion to the revised manuscript. Please see Page 2 in the revised manuscript.

2. The proposed co-doping of $\text{Fe}(\text{TOF})^+$ into PANI is not convincing, as reflected by Fig. 3(d). For Fe-free electrolytes, there are still distinct CV profiles present; while in $\text{Fe}(\text{TOF})^+$ cases, readers still notice weak but very similar current signals. The current response proves the cathode performance is highly correlated with the proton concentration rather than ferruginous ion species.

Response:

Thank you for your insightful comment. We agree protons contribute remarkably to the capacity. On the other hand, to present a detailed mechanism study, we performed Raman spectrum measurements of electrolytes with various concentrations and the powder of $\text{Fe}(\text{TOF})_2$ to clarify the possible Fe^{2+} solvated structure in the 1.0 M $\text{Fe}(\text{TOF})_2$ electrolyte, which can affect charging and discharging process of the battery (Fig. R1-1).

Fig. R1-1 Raman spectra of H_2O , $\text{Fe}(\text{TOF})_2$ powder and $\text{Fe}(\text{TOF})_2$ electrolytes with different concentrations.

Fig. R1-2 Raman spectra of H-bond in $\text{Fe}(\text{TOF})_2$ solutions with different concentrations.

Fig. R1-3 Raman spectra of O=S bond of $\text{Fe}(\text{TOF})_2$ electrolytes with different concentrations and $\text{Fe}(\text{TOF})_2$ powder.

As shown in Fig. R1-2, after increasing the $\text{Fe}(\text{TOF})_2$ concentration from 0.5 M to 1 M, the strong hydrogen bond at 3555 cm^{-1} becomes more intensive, indicating that more TOF^- ions participate in the coordination shell of the cation in the solvation structure. Additionally, the Raman characteristic peak at 1034 cm^{-1} , belonging to the vibration of O=S bond of TOF^- , can be deconvoluted to free OTF (O=S signal at 1032 cm^{-1}) and contact ion pairs (CIP, Fe-O=S signal at 1038 cm^{-1}). Compared to the 0.5 M

Fe(TOF)₂ electrolyte, more Fe-O=S appeared in the 1 M Fe (TOF)₂, indicating the OTF⁻ anions participate in the solvated structure of Fe²⁺ in the 1 M Fe (TOF)₂ electrolyte (Fig. R1-3). The existence of this solvated structure containing TOF⁻ indicates the Fe(TOF)⁺ complex ions may co-participate in the reaction with C-PANI during the discharge process.

Fig. R1-4 FT-IR spectra before and after polishing the C-PANI electrodes at full discharge/full charge

To further confirm this conjecture, the C-PANI cathode material was characterized at the fully charged state and fully discharged state by FT-IR. A very broad O=S vibrational peak appears near 1030 cm⁻¹ at the discharge state, suggesting that the chemical environment of O=S adsorbed on the C-PANI surface has been changed at a fully discharged state. The broad peak contains both Fe(TOF)₂ adsorbed on the surface and Fe(TOF)⁺ reacted with C-PANI. After polishing the C-PANI electrode, a shoulder peak appears at around 1022 cm⁻¹ due to the alteration of the O=S chemical environment, unraveling the co-uptake of TOF⁻ and Fe²⁺ and the reaction with C-PANI. In the following fully charged state, the O=S peak is present at 1030 cm⁻¹ before and after polishing, while it is never presented at 1021 cm⁻¹, suggesting that the broad 1030 cm⁻¹ is likely from Fe(TOF)₂ adsorbed on the surface.

Fig. R1-5 XPS spectra of Fe 2p of C-PANI cathodes obtained at the state of discharging in different etching time.

Additionally, the mechanism of $\text{Fe}(\text{TOF})^+$ participation in the reaction was further analyzed by XPS. The high resolution Fe 2p XPS spectrum of full-discharged C-PANI shows five peaks at around 712.3 eV, 715.1 eV (Fe^{2+}), 720 eV (satellite) and 726 eV, 730.1 eV (Fe^{2+}) correlating with the inevitable adsorption of $\text{Fe}(\text{TOF})_2$ on the surface of C-PANI. After 2 min etching, the two new peaks at 710.0 eV and 724.2 eV emerged and got intensified with the etching time being extended to 4 minutes,

implying Fe^{2+} coordinates with N of C-PANI and forming Fe-N structure after fully discharged.

Fig. R1-6 XPS spectra of S 2p of C-PANI cathodes obtained at the state of discharging in different etching time and $\text{Fe}(\text{TOF})_2$.

Meanwhile, the high resolution S 2p spectra also showed a noticeable variation after discharge. As shown in Supplementary Figure 23 or Fig.R1-6, three peaks at 169 eV, 167.5 eV and 162.5 eV, which didn't present for the C-PANI at full discharged state, began to appear and intensify with increasing etching time. The most reliable explanation is the coordination between Fe^{2+} and TOF forming $\text{Fe}(\text{TOF})^+$ complex ions, which reacts with N in the discharged state, leading to a dramatic change in the S chemical environment. The coordination mode is shown in the ball-and-stick structure (inset of Fig. R1-6)

We are sorry for having not made this point clear. Please see pages 12-13 in the revised manuscript and Pages 15-20 (Supplementary Figure 18-23) in the revised supplementary information. We have also added the corresponding references.

3.The in-situ Raman detections (Fig. 3(b)) neither show any direct evidences. Though ex-situ XPS spectra reveal the chemical state change of Fe on PANI surfaces, the bulk insertion of $\text{Fe}(\text{TOF})^+$ is still unknown.

Response:

Thanks for your insightful comments. We supplemented the XPS and FT-IR of full discharge/full charge state of C-PANI to further explain the formation of complex $\text{Fe}(\text{TOF})^+$ and $\text{Fe}(\text{TOF})^+$ ions co-participating in the reactions during the discharge process. For example, in FT-IR spectra, a very broad S=O vibrational peak appears near 1030 cm^{-1} at the discharge state, suggesting that the chemical environment of S=O adsorbed on the C-PANI surface has been changed at fully discharged. The broad peak contains both $\text{Fe}(\text{TOF})_2$ adsorbed on the surface and $\text{Fe}(\text{TOF})^+$ reacted with C-PANI. After polishing the C-PANI electrode, a shoulder peak appears around 1022 cm^{-1} due to alteration of the S=O chemical environment, unraveling the co-uptake of TOF^- and Fe^{2+} and reaction with C-PANI. In the following fully charged state, the S=O peak is present in 1030 cm^{-1} before and after polishing, while it is never present in 1021 cm^{-1} , suggesting that the broad 1030 cm^{-1} is from $\text{Fe}(\text{TOF})_2$ adsorbed on the surface.

In XPS results, the high resolution Fe 2p XPS spectra of full-discharged C-PANI show five peaks at around 712.3 eV, 715.1 eV (Fe^{2+}), 720 eV (satellite) and 726 eV, 730.1 eV (Fe^{2+}) correlating with the inevitable adsorption of $\text{Fe}(\text{TOF})_2$ on the surface of C-PANI. After 2 min etching, the two new peaks at 710.0 eV and 724.2 eV emerged and get intensified with the etching time being extended to 4 minutes, implying Fe^{2+} coordinates with N of C-PANI and forming Fe-N structure after fully discharged. In the meantime, the high resolution S 2p spectra also showed a noticeable variation after discharge. As shown in Supplementary Figure S23, three peaks at 169 eV, 167.5 eV and 162.5 eV, which were not present on the C-PANI at full discharged state, begin to appear and intensify with increased etching time. The explanation is the formation of coordination between Fe^{2+} and TOF to form $\text{Fe}(\text{TOF})^+$ complex ions, which reacts with N at the discharged state, leading to a dramatic change in the S chemical environment. The coordination mode is shown in the ball-and-stick structure (inset of Fig. R1-6).

We have added a related discussion to the revised manuscript. Please see pages 12-13 in the revised manuscript and Pages 15-20 (Supplementary Figure S18-23) in the revised supplementary information. We have also added the corresponding references.

4. Authors neglect significant research on the Fe anode side. Though using proton-rich solution (pH: ~ 0.5) benefits the redox reactions of PANI, the use of highly acidic

electrolyte is quite detrimental to Fe anodes, leading to parasitic reactions like severe metallic corrosion, the build-up of thick interfacial passivation layers, etc.

Response:

Thanks for the insightful comment. Following the reviewer's suggestion, we have conducted a series of experiment to investigate the Fe anode, including Fe||Fe symmetric cells, electrochemical impedance spectroscopy at different cycles, coulombic efficiency of Cu||Fe cells, SEM of Fe||Fe symmetric cells and coulombic efficiency of Cu||Fe cells.

Anions have an important impact on the de-solvation process. The CF_3SO_3^- of 1M Fe(TOF)₂ possesses weaker interactions with H₂O/metal ions than other anions (SO_4^{2-} , NO_3^- , Cl^-) in aqueous batteries (*ACS Energy Lett.* 2021, 6, 2704–2712), which can further be favorable for the de-solvation process, resulting in fast kinetics and high coulombic efficiency. We then evaluated the symmetric cells assembled. The charge/discharge test (Fig. R1-7) was conducted under a current density of 1 mA/cm². The potential polarization is over ~ 640 mV and can run steadily over 360 h without evident potential fluctuations.

Fig. R1-7 The voltage profiles of Fe||Fe at a current density of 1 mA cm⁻² and capacity of 1 mA h cm⁻².

Fig. R1-8 EIS of the Fe||Fe symmetric cells at different cycles.

The evolution of the impedance results of the symmetric cells revealed their electrochemical stability (Fig. R1-8). The R_{ct} of the Fe cell first increased and then decreased, eventually reached steady state after 96 hours. Such large variations in the R_{ct} of the Fe cell indicate the enlarged surface areas and other inactive products during the cycled process. The potential polarization of the first cycle (0.69 V) is greater than that of the following cycles, due to higher nucleation overpotential in the initial interval (Fig. R1-9a). Furthermore, the morphologies of Fe electrode were further investigated after plating and tripping for 1 hour at 1 mA h cm^{-2} . The plated Fe exhibited nonuniform Fe morphology with small Fe granules. Also, the striped Fe shows the shape of pits and concave (Fig. R1-9 b, c). After 168 hours, the surface of the plated iron is dense and composed of large particles of a few micrometers in size under scanning electron microscopy (Fig. R1-10). The large-scale morphology is beneficial for improving the stability of the battery. (*Energy Storage Materials*, 2022, 52, 329-354).

Fig. R1-9 (a) GCD plots of a Fe||Fe cell at different cycle numbers. (b) SEM images of Fe electrode before cycling. (c) Fe electrode after 1-hour electroplating. (d)

Fe electrode after 1-hour stripping.

Fig. R1-10 SEM images of Fe electrode after 168 hours.

We have added related discussion into the revised manuscript. Please see Page 7-9 (Supplementary Figure 6-10) in the revised Supplementary Information. We have also added the corresponding references.

We agree with reviewers that the issue of the Fe anode is important in aqueous batteries. The focus of this research is to develop a highly stable cathode material. Aqueous iron-ion batteries are in the early stage of development, there is a lack of high-performance Fe-bearing cathode materials coupled to ferrous metal anodes.

The development of aqueous iron-metal-based batteries has a long history. Some artisans in ancient Iraq may have invented Baghdad batteries with iron metal anode as early as 200BC. The formal rise of iron batteries began with the nickel-iron alkaline batteries invented by Edison in 1901. Subsequently, iron-air batteries and iron redox flow batteries developed in succession. But the research on iron batteries seems interrupted after 1980 since the lead-acid batteries and Li-ion batteries emerged.

Nevertheless, renewed attention has been paid to iron batteries recently due to the consideration of low cost and environmental friendliness. More recently (2019), Ji et al. proposed a novel iron battery with slightly acidic electrolyte, which brings the insight that iron is an undervalued anode candidate in mild/slightly acidic aqueous batteries, and a series of novel iron batteries with V_2O_5 , S/C, and $I_2||N$ -HPC as cathode have been developed. However, the results of inferior cycle life (typically <2000) and low specific capacity are still unsatisfactory and under expectations.

5. The super-long 39,000-cycle battery lifespan (coulombic efficiency always approaches 100 %; Fig. 2f) is surprising but hardly understood, since the Fe electroplating/stripping is not an ideally reversible process (actually, there are inevitable side reactions occurring, particularly at the anode side). So, how to explain this result?

Response:

Thank you for your insightful comments. The charge/discharge profiles at various cycles show in Figure R1-11. To double confirm our stability, we have tested another sample, and similar performance were obtained at a loading mass of $\sim 1.1 \text{ mg cm}^2$ in Figure R1-12.

Fig. R1-11 The discharging - charging profiles of C-PANI based cell showing in Figure 2f.

Fig. R1-12 (a) The long-cycle profiles for another Fe||C-PANI cell (loading mass of $\sim 1.1 \text{ mg cm}^2$) at 25 A g^{-1} . (b) The discharging - charging profiles of the cell.

To check the Fe chemical stability, the Coulombic efficiency of metal plating/stripping was conducted in asymmetrical Fe||Cu cells. As shown in Fig. R1-13, The Coulombic efficiency is $\approx 37.2\%$ in the first cycle, possibly attributing to unstable anode interface reactions like passivation layer fracture or reformation, and then the Coulombic efficiency reach $\sim 100\%$ in subsequent cycles. Additionally, the morphology of iron deposited for 1 hour were investigated simultaneously. The plated iron exhibited smooth submicrometer-sized particles and compactly stacked, which facilitates improved battery performance. Practical metal anode based batteries use an excess mass of Fe metal, which is also the reason of highly stability even in the presence of side reactions.

Fig. R1-13 (a) Coulombic efficiency of an asymmetrical Fe||Cu coin cell. (b) An SEM image of the plated Fe metal in the first cycle. (c) A plating/stripping efficiency test.

Moreover, there are many cells reported with ultra-long cycle in the field of aqueous batteries, even though the electroplating/stripping of the metal electrode is not an ideally reversible process. For example, the Coulombic efficiency of 100 % phenomenon also was reported in the *J. Am. Chem. Soc.* 2022, 144, 25, 11444–11455 (Aqueous Al Battery), *J. Mater. Chem. A*, 2022,10, 4739-4748 (Aqueous Al Battery), *Adv. Mater.* 2021, 33, 2105234 (Aqueous Fe Battery), *Energy Storage Materials* 28 (2020) 247–254 (Aqueous Fe Battery), *Small Methods* 2021, 5, 2100611 (Aqueous Al Battery), *Nature communications* 2022, 13, 576 (Aqueous Al Battery), *Adv. Funct. Mater.* 2021, 31, 2102063 (Aqueous Al Battery). The anodes of these batteries also undergo side reactions, especially in aqueous Al batteries.

The reason of the above-mentioned batteries can achieve high Coulombic efficiencies can be briefly summarized as follows:

1. The anode metals of the aqueous batteries reported above are all in mass excess, even if there are side reactions, and are still in excess during charging and discharging;
2. Protons possess small radii, and fast diffusion in aqueous electrolytes, and participate in the reaction during the charging/discharging process;

In addition, in our batteries, the mixed-cations $\text{Fe}(\text{TOF})^+$ with shielding effect reduce the strong electrostatic interaction with host materials, leading to fast reaction kinetics and enhanced electrochemical performance. We further confirmed the reliability of the C-PANI with another battery that showed superior cycling stability for 39700 cycles.

We have added related discussion to the revised supplementary information. Please see the revised supplementary information Figure 11-13 (supplementary information page 9-11).

6. Likewise, the superb rate capability (120 mAh g^{-1} retained under 25 A g^{-1}) is also confusing. How can authors account for such outstanding rate performance of battery devices built by conducting polymers with less electronic conductivity than carbons? High mass loading is required to demonstrate the high-rate performance.

Response:

Thank you for your valuable comment. Following your constructive suggestion, we added the rate performance data of high mass loading in Fig. R1-14-15 ($\sim 3.5 \text{ mg cm}^2$ and $\sim 8.0 \text{ mg cm}^2$). Actually, with the rapid progress achieved in nanoscience and nanotechnologies, the boundary between battery material and capacitive material (particularly pseudocapacitive materials) has been becoming blurred in recent years. There is debate on how to accurately distinguish these two types of materials (*ACS Nano* 2018, 12, 3, 2081-2083; *Small*, 2020, 16 (37), 2002806; *Energy Environ. Mater.*, 2019, 2 (1), 30-37). However, the latest view is fortunate that it is a continuum between double-layer capacitance and Faradaic intercalation (*Nature Energy*, 7, 3, 222-228). According to this view, pseudocapacitance is a result of both capacity contribution and battery behavior. Then pseudocapacitance usually brings about the high rate performance of electrode.

Fig. R1-14 (a) the rate capability of Fe||C-PANI cells (a loading mass of $\sim 3.5 \text{ mg cm}^{-2}$) at various current densities from 1 to 25 A g^{-1} and (b) corresponding charge/discharge profiles.

Fig. R1-15 (a) the Rate capability of Fe||C-PANI cells (a loading mass of $\sim 8 \text{ mg cm}^{-2}$) at various current densities from 1 to 25 A g^{-1} and (b) corresponding charge/discharge profiles.

Although carbon cloth is more conductive than C-PANI, carbon cloth only delivered a capacity of 7 mAh in the voltage range from 0 to 1.3 V, which is also illustrated in our manuscript (Supplementary Figure 11). Moreover, the charge storage of C-PANI is based on a bipolar-type redox reaction. Not only anions can interact with the positively charged nitrogen atoms ($\text{C}=\text{N}^+-$) in the oxidized PANI, but also cations can be stored in electronegative nitrogen sites ($-\text{N}^-$) of reduced PANI during the discharge process.

We have added the new data in the revised supplementary information (page 12-13).

7. The charge/discharge voltage profiles (Fig.2b) are inconsistent to CV plots (Fig.2a). The potential plateaus become slope-like profiles. Note the voltage profiles are seemingly close to a triangle shape, as often recorded in pseudocapacitor testing. Also, a huge voltage polarization is noticed evidently.

Response:

Thank you for your valuable and constructive comment. I am very sorry for the improper expressions that led to your misunderstanding. The “consistent” means only that the voltage plateau is consistent. For the Fe||C-PANI battery, there are two reduction peaks at 0.9 and 0.56 V, as well as two reduction peaks at the relatively lower voltages of 0.78 and 0.39 V (Fig. 2a). These expressions have been revised for better clarity. Please see highlighted parts on Page 7 in the revised manuscript.

It is an indisputable fact that high current densities induce large polarization, and such a phenomenon has been widely reported in *Energy Storage Materials*, 2022, 44, 517–526, *J. Am. Chem. Soc.* 2021, 143, 15369–15377, *Angew. Chem. Int. Ed.* 2021, 60, 20826–20832, *Angew. Chem. Int. Ed.* 2022, 61, 42, 61, e202211107, *Adv. Mater.* 2021, 33, 2105234, *Adv. Mater.* 2022, 34, 2206963. The potential plateaus also become slopy profiles.

Fig. R1-16 (a) Zn||PANI batteries (*Energy Storage Materials*, 2022, 44, 517–526), (b) Zn||poly(1,5-naphthalenediamine) batteries (*J. Am. Chem. Soc.* 2021, 143, 15369–15377). (c) Zn||bis(phenylamino)phenothiazine-5-ium iodide (PTD-1) (*Angew. Chem. Int. Ed.* 2021, 60, 20826–20832). (d) Zn||poly(1,8-diaminonaphthalene) (PDAN) (*Angew. Chem. Int. Ed.* 2022, 61, 42, 61, e202211107). (e) Fe||VOPO₄·2H₂O batteries (*Adv. Mater.* 2021, 33, 2105234). (f) Zn||PANI batteries (*Adv. Mater.* 2022, 34, 2206963.).

Truly as the reviewer said, this C-PANI material exhibits a supercapacitor-level high-rate capability, which is also explicated in our manuscript. Then pseudocapacitance usually brings about the electrode’s high-rate performance. Typically, capacitive materials basically exhibit triangle shape voltage profiles, while battery-type materials show obvious CV peaks and detectable charge/discharge plateaus at the current density of 5 A g⁻¹. The battery presents obvious discharge

plateaus at around 0.90 V and 0.4 V, although these discharge plateaus becoming indistinct at high rates. So, the C-PANI material is considered as a battery-type cathode material despite its capacitor-level high-rate capability.

We have added related discussion into the revised manuscript. Please see Page 7-8 in the revised manuscript.

Reviewer #2 (Remarks to the Author):

The authors prepared highly conductive cross-linked PANI (C-PANI) using melamine as the crosslinker and applied them as electrodes for aqueous iron batteries. The Fe//C-PANI batteries exhibited fast reaction kinetics and record-breaking battery performance including super-high rate capability (120 mAh g^{-1} at 25 A g^{-1}) and a super-long 39000 cycle life, which represents a solid advance on aqueous Fe batteries. More importantly, the proton- $\text{Fe}(\text{TOF})^+$ co-storage mechanism was demonstrated. Multifunctional Fe//C-PANI electrochromic batteries were also developed to integrate both electrochromism in the visible and middle-Infrared and energy storage, demonstrating great potential in next-generation smart battery technologies. The experimental results are convincing and are analyzed in sufficient depth. Therefore, I recommend that this work can be published in Nature Communications. And some revisions are needed to improve the manuscript further before the acceptance.

1. Why choosing 1 M $\text{Fe}(\text{TOF})_2$ as the electrolyte? FeSO_4 is a commonly used electrolyte, and it can be much cheaper. Since one of the merits of Fe batteries is its low cost, the authors should explain their choice on electrolyte more profoundly.

Response:

We appreciate your insightful comments. We also agree that the goal of scientific research should be practicality and there is a long way to go for practical application of the developed batteries. The reasons for choosing $\text{Fe}(\text{TOF})_2$ as the electrolyte can be explained from both scientific and technological perspectives.

Scientifically, anions have an important impact on the de-solvation process. The CF_3SO_3^- possesses weaker interactions with H_2O /metal ions than other anions (SO_4^{2-} , NO_3^- , Cl^-) in aqueous batteries (*ACS Energy Lett.* 2021, 6, 2704–2712), which can further be favorable for the de-solvation process, resulting in the fast kinetics, and high coulombic efficiency.

In addition, the pH of the FeSO₄ electrolyte will change significantly due to reaction between H⁺ with C-PANI during the charging and discharging process, leading to the formation of Fe(OH)₂ precipitate and a decrease in stability.

Technologically, we admit that, in terms of technology maturity, iron batteries are still much behind lithium-ion batteries. There will be many uncertain factors to realize practical applications. Aqueous iron batteries are at an early stage of development. The development of aqueous iron batteries have a long way to go. If iron batteries are commercialized in the future, the price of Fe(TOF)₂ may be reduced as commercialization progresses, or it may be replaced by other iron salts with better electrochemical properties.

We have added a related discussion to the revised manuscript. Please see the highlighted part on page 7 in the revised manuscript.

2. The standard electrode potentials of the Fe³⁺/Fe²⁺ and Fe²⁺/Fe couples are +0.77 and -0.44 V versus SHE, respectively, which gives rise to a theoretical electrochemical window of ≈1.21 V for a Fe metal battery. However, why did the author charge the battery to 1.3 V? Is there any problem with that?

Response:

Thank you for your valuable comment. Although the theoretical electrochemical window is 1.21 V in our experiments, the oxidation peak from divalent iron to trivalent iron ions still does not appear even when the cell is charged to 1.3 V. Otherwise, the Fe³⁺ hydrolysis and the consequent Fe(OH)₃ precipitation would take place, resulting in inferior cycle life. The previous reports have shown that the Fe plating occurs at the potential of -0.64 V versus SHE (-0.20 V vs Fe²⁺/Fe), which is -0.32 V lower than the HER thermodynamic potential (Adv. Funct. Mater. 2019, 29, 1900911). So, the electrochemical window in our battery can be above 1.3 V.

We have added related discussion into the revised manuscript. Please see the highlighted part in page 7 in the revised manuscript.

3. In Fig. 3d, are the CV results for the same electrode sample? If yes, please clarify; if not, please express the vertical axis units in terms of current density.

Response:

We regret for this oversight and thank you for your suggestion. We have revised it in our manuscript. Please see the highlighted part on page 9 in the revised manuscript

4. Why gold is coated on the surface of nylon 66? Is it okay if it is another metal? Please further clarify this point.

Response:

Thank you for your valuable comment. The gold is used in our research for three reasons: 1. electrochemical stability; 2. high electrical conductivity; 3. high IR reflectivity, this is most important for IR electrochromic. We have added related discussion into the revised manuscript. Please see the highlighted part in page 14 in the revised manuscript. And we have supplemented the infrared reflectance spectra of Au-coated nylon66 in Supplementary Figure 26.

Fig. R2-1 Specular FTIR diffuse reflectance of Au-coated nylon66.

5. The forward and backward peaks in CV curves should be well assigned for better clarity (Figure 5a), the reactions should be well marked.

Response:

We thank the referee for the valuable comment. These forward and backward peaks in CV curves have been marked. Please see Fig. 5a in the revised manuscript.

6. Why are there some rise and fall in the long cycle test in Fig. 2f?

Response:

Thank you for the valuable comment. This is relevant to the alteration of the thermodynamic kinetics of the electrolyte caused by fluctuations in temperature during the day and night, which eventually result in fluctuating capacity. This common phenomenon also appears in previous literature. (Angew. chem. int. ed. 2022, e202210979, doi:10.1002/ange.202210979, *Adv. Energy Mater.* **2019**, 9, 1900993, *Advanced Materials*, DOI: 10.1002/adma.202206754)

We have added the information into the revised manuscript. Please see the

highlighted part in page 8 in the revised manuscript.

7. For the TEM observation of C-PANI, how to peel it off from the substrate?

Response:

Thank you for the valuable comment. The C-PANI for the TEM was scraped off the carbon cloth with a knife.

We have added related explanations into the revised manuscript. Please see the highlighted part on page 19.

Reviewer #3 (Remarks to the Author):

In this article, the author developed an aqueous iron battery with long cycle life and high rate capability. Given its innovative and excellent electrochemical properties, I think it could be published in Nature communications. Aqueous iron-ion batteries are in the early stage of development, there is a lack of high-performance Fe-bearing cathode materials coupled to ferrous metal anodes. The battery of iron metal has poor cycle stability and low Coulombic efficiency. The C-PANI cathode used in this paper has not been explored in previous iron batteries, which is relatively innovative. The developed battery has an extraordinary cycle life of 39,000 cycles, high rate capability, and an initial specific capacity of 209 mAh/g at 5 A/g, with excellent electrochemical performance. But there is still room for improvement in this paper, details as follows.

1. Isn't the vibration of the -NH= structure at 1139cm^{-1} in Fig. 1b?

Response:

We regret for this oversight. It should be the vibration of the δ (C-H). It has been corrected.

Please see highlighted parts on Page 5 in the revised manuscript. We have also double checked our manuscript with references.

2. Figure 2a can be more standardized, and the position and scanning direction of the redox peak can be marked.

Response:

Thanks for detailed comments. Following your comment, we have revised it in our manuscript. Please see Fig. 2a on Page 6 in the revised manuscript.

3. Page 9, line 14, 1476 cm^{-1} (C=N) The shoulder strap at 1488 cm^{-1} appeared only after the first charge to 0.9 V.

Response:

Thank you for your valuable comment. The shoulder strap at 1488 cm^{-1} is attributed to the coordination of Fe^{2+} with imine nitrogen of semi-oxidized state. The reaction of polyaniline with hydrogen ions occurs before the reaction with Fe^{2+} in the first cycle of the discharge process. The appearance of the shoulder strap at 1488 cm^{-1} is accompanied by a substitution reaction of iron ions in C-PANI by hydrogen ions (Fig. 3i). So, the shoulder strap appeared only after the first charge to 0.9 V.

We have added related discussion to the revised manuscript. Please see the highlighted part in page 10 in the revised manuscript.

4. What is the irreversible peak at 1.3 V in Fig. 5a assigned for?

Response:

Thank you for your comment. It has been previously reported (Res. 17), the irreversible peak at about 1.3 V (should be slightly $>1.3\text{V}$) is the electrochemical overoxidized and degradation process (Electrochimica Acta, 2006, 52, 234–239; J solid state electrochem, 1998, 2, 355-361). The final degradation products consist of p-benzo-quinone, hydroquinone, p-aminophenol, quinoneimine. The following is electrochemical overoxidized and degradation mechanism (Fig. R3-1):

Fig. R3-1 PANI degradation mechanism (Electrochimica Acta, 2006, 52, 234–239; J solid state electrochem, 1998, 2, 355-361).

We have added related discussion into the revised manuscript. Please see the highlighted part in page 16 in the revised manuscript.

5. Figure 5d, it can be seen from the figure that the Coulomb efficiency of the battery has decreased significantly after 20,000 cycles. What is the specific reason? Why the Coulomb efficiency value fluctuated with fixed step?

Response:

Thank you for your valuable and constructive comment. This phenomenon is caused by the following two reasons. 1. The flexible electrochromic batteries are not as well sealed as coin-type cells, and after 20000 cycles the hydrogel electrolyte gradually loses its water content. 2. A possible reason for the gradual decrease in coulomb efficiency is the electrochemical degradation of C-PANI, and the gradual formation of small molecule degradation products (J solid state electrochem, 1998, 2, 355-361).

We have added related discussion into the revised manuscript. Please see the highlighted part on page 16 in the revised manuscript.

6. The detailed Fe storage mechanism should be investigated. XPS spectra of Fe should be included for Fe-stored PANI.

Response:

Thank you for your valuable and constructive comment. Following your suggestion, we further reinforce the investigation and explanation of XPS and FT-IR results.

Fig. R3-2 XPS spectra of Fe 2p of C-PANI cathodes obtained at the state of discharging in different etching time.

Fig. R3-3 XPS spectra of S 1s of C-PANI cathodes obtained at the state of discharging in different etching time and $\text{Fe}(\text{TOF})_2$.

The high-resolution Fe 2p XPS spectrum of full-discharged C-PANI shows five peaks at around 712.3 eV, 715.1 eV (Fe^{2+}), 720 eV (satellite) and 726 eV, 730.1 eV (Fe^{2+}) correlating with the inevitable adsorption of $\text{Fe}(\text{TOF})_2$ on the surface of C-PANI. However, after 2 min etching, the two new peaks at 710.0 eV and 724.2 eV emerged and got intensified with etching time extended to 4 minutes, implying Fe^{2+} coordinates with N of C-PANI and forms Fe-N structure after full discharge (Fig. R3-2). In the meantime, the high resolution of S 2p also showed a noticeable variation after discharge. As shown in Supplementary Figure 23, three peaks at 169 eV, 167.5 eV and 162.5 eV, which were not presented for the C-PANI at full discharged state, appeared and intensified with increasing etching time. The most likely explanation is the $\text{Fe}(\text{TOF})^+$ complex ion formation of coordination between Fe^{2+} and TOF, which reacts with N in the discharged state, leading to a dramatic change of the S chemical environment. The coordination mode is shown in the ball-and-stick structure (inset of Supplementary Figure S23).

Fig. R3-4 FT-IR spectra before and after polishing the C-PANI electrodes at full discharge/full charge.

Additionally, FT-IR spectra have also been used to study Fe^{2+} store mechanisms. As shown in Fig. R3-4, a very broad S=O vibrational peak appears near 1030 cm^{-1} at the discharge state, suggesting that the chemical environment of S=O adsorbed on the C-PANI surface has been affected. The broad peak at this point contains both $\text{Fe}(\text{TOF})_2$ adsorbed on the surface and $\text{Fe}(\text{TOF})^+$ reacted with C-PANI. After polishing the C-PANI electrode, a shoulder peak appears around 1022 cm^{-1} , unraveling the co-uptake of TOF^- and Fe^{2+} and reaction with C-PANI. In the following fully charged state, the S=O peak is present in 1030 cm^{-1} before and after polishing, while it is never present in 1021 cm^{-1} , suggesting that the broad 1030 cm^{-1} is from $\text{Fe}(\text{TOF})_2$ adsorbed on the surface.

We are sorry for having not made this point clear. We have added related discussion into the revised manuscript. Please see Pages 12-13 in the revised manuscript and Pages 15-20 (Supplementary Figure S18-23) in the revised supplementary information. We have also added the corresponding references.

REVIEWERS' COMMENTS

Reviewer #1 (Remarks to the Author):

I have read carefully the comments from all the referees and the author's response. I think the authors have made considerable changes to the manuscript and the current version can be considered for acceptance after the following minor revisions:

- 1, There is a solid line in Figure 5, which should be deleted.
- 2, It is suggested to include the current density value in Figure 1a and b.

Reviewer #2 (Remarks to the Author):

The authors have made sufficient supplementary results and discussion according to the reviewers' suggestions. The results are more convincing in the present version, and I believe that this contribution can give inspirations for the Fe-based aqueous batteries.

Reviewer #3 (Remarks to the Author):

The authors have properly revised the manuscript. I recommend it's publication in Nature Communications.

Response to reviewers' comments

Reviewer #1:

I have read carefully the comments from all the referees and the author's response. I think the authors have made considerable changes to the manuscript and the current version can be considered for acceptance after the following minor revisions:

1, There is a solid line in Figure 5, which should be deleted.

Response: Thanks very much for your positive comments.

Following your suggestion, we have deleted the solid line in Figure 5 in the revised manuscript.

2, It is suggested to include the current density value in Figure 1a and b.

Response: Thanks very much for your positive comments.

Following your suggestion, we have added the current density value in Figure 2a and 2b in the revised manuscript.

Reviewer #2:

The authors have made sufficient supplementary results and discussion according to the reviewers' suggestions. The results are more convincing in the present version, and I believe that this contribution can give inspirations for the Fe-based aqueous batteries.

Response: Thanks very much for your positive comments.

Reviewer #3 (Remarks to the Author):

The authors have properly revised the manuscript. I recommend its publication in Nature Communications.

Response: Thanks very much for your positive comments.